# Assessment of climate change impact and difference on the river runoff in four basins in China under 1.5 ℃ and 2.0 ℃ global warming

Hongmei Xu[1], Lüliu Liu[1], Yong Wang[2], Sheng Wang[3], Ying Hao[4], Jingjin Ma[5], Tong Jiang[6,1]

[1]National Climate Center, China Meteorological Administration, Beijing, 100081, China
[2]Chongqing Climate Center, Chongqing, 401147, China
[3]Anhui Climate Center, Hefei, 230031, China
[4]Anhui Meteorological Observatory, Hefei, 230031, China
[5]Beijing Meteorological Disaster Prevention Center, Beijing, 100089, China
[6]Collaborative Innovation Center on Forecast and Evaluation of Meteorological Disasters, School of Geography and Remote Sensing, Nanjing University of Information Science & Technology, Nanjing, 210044, China

*Correspondence to*: Hongmei Xu (xuhm@cma.gov.cn)

*Co-correspondence to:* Tong Jiang (jiangtong@cma.gov.cn)

**Abstract:** To quantify climate change impact and difference on basin-scale river runoff under the limiting global warming thresholds of 1.5 ℃ and 2.0 ℃, this study examined four river basins covering a wide hydroclimatic setting. We analyzed projected climate change in four basins, quantified climate change impact on annual and seasonal runoff based on the Soil Water Assessment Tool, and estimated the uncertainty constrained by the global circulation models (GCMs) structure and the Representative Concentration Pathways (RCPs). All statistics for the two river basins (the Shiyang River (SYR) and the

Chaobai River (CBR)) located in northern China indicated generally warmer and wetter conditions, whereas the two river basins (The Huaihe River (HHR) and the Fujiang River (FJR)) located in southern China projected less warming and were inconsistent regarding annual precipitation change. The simulated changes in annual runoff were complex; however, there was no shift in seasonal runoff pattern. The 0.5 ℃ global warming difference resulted 0.7 ℃ and 0.6 ℃ warming in basins in northern and southern China, respectively. This led to projected precipitation increase by about 2% for the four basins, and

to a decrease in simulated annual runoff of 8% and 1% in the SYR and the HHR, respectively, but to an increase of 4% in the CBR and the FJR. The uncertainty in projected annual temperature was dominated by the GCMs or the RCPs; however, that of precipitation was constrained mainly by the GCM. The 0.5 ℃ difference d

ecreased the uncertainty both in the annual precipitation projection and the annual and monthly runoff simulation.

Keywords: 1.5 ℃ and 2.0 ℃ warming; runoff; Shiyang River; Chaobai River; Huaihe River; Fujiang River

# 1 Introduction

In addition to changes in other variables of the climate system, global temperature has shown warming of 0.85 ℃ during 1880–2012 and further increase of 2.0–4.0 ℃ is projected over the next 100 years (IPCC, 2013). The goal of 1.5 ℃ and 2.0 ℃ global warming relative to the preindustrial climate has been proposed to avoid the dangerous effects of anthropogenic climate change (UNFCCC, 2015). The observed changes in climate have affected both natural and human systems in recent decades. The level of climate change risk at 1.0 ℃ or 2.0 ℃ global warming is thought considerable, while that associated with an increase of ≥4.0 °C global warming is considered high to very high (IPCC, 2014). Significant progress has been achieved in comprehensive quantitative assessments of aggregate global climate impact (Schellnhuber et al., 2014). However, climate research is also challenged to provide more robust information on the impact of climate change under different scenarios of global warming (particularly at local and regional scales) to assist the development of sound scientific adaptation and mitigation measures (Huber et al., 2014). For example, a number of areas have been identified with severe projected impacts of warming at 2.0 ℃ (Schleussner et al., 2016).

Observed climate change has caused changes in global hydrological cycle, and this is expected to have considerable impact on multiple scale freshwater availability (Schmied et al., 2016). Most regional changes in precipitation can be attributed either to internal variability of the atmospheric circulation or to global warming. Climate change over the 21$^{st}$ century is projected to reduce renewable surface water significantly in most dry subtropical regions, while water resources are projected to increase at high latitudes (IPCC, 2014). At the global scale, the extreme rainfall is projected to more frequency under both 1.5 ℃ and 2.0 ℃ warming until around 2070s; however, the increase is expected to be higher under 2 ℃ warming after the late 2030s (Zhang and Villarini, 2017). Furthermore, global warming of 2.0 ℃ is anticipated to affect natural runoff in river basins around the world and to dominate runoff changes, even considering human impact (Haddeland et al., 2014). Global warming of 2.0 ℃ will enhance water scarcity in areas projected to experience severe water resources reduction, although uncertainties exist in the projected changes in discharge and in the spatial heterogeneity depending on the contributions from global hydrological models and global climate models (Schewe et al., 2014). For the most region with simulated water resource decrease, the uncertainties in simulated runoff usually constrained by global hydrological models, which suggests the necessity for improvement of regional- or local-scale hydrological projections (Su et al., 2017). Comparison of the performance of global and regional hydrological models indicates that regional hydrological models are better able to represent the long-term average seasonal dynamics (Hattermann et al., 2017; Gosling et al., 2017).

Within the context of the global temperature increase, China has experienced robust warming that is characterized by the greatest rate of annual mean temperature increase (i.e., more than 0.3 ℃/10a during 1961–2012) in northern areas (Third National Assessment Report for Climate Change, 2015). River runoff has decreased consistently in the Yellow, Liao, and Songhua rivers but increased in the Pearl River because of increased precipitation in southern China and decreased precipitation in northern China combined with human activities (Xu et al., 2010a). The runoff of rivers located in northern China, in areas with arid and semiarid climate, is more sensitive to precipitation than in southern China (Xie et al., 2018).

The 2.0 ℃ warming threshold will be exceeded under two Representative Concentration Pathways (RCPs), averaged across China, will be around 2033 ± 15a under RCP4.5 and 2029 ± 10a under RCP8.5 (Chen and Zhou, 2016). Simulations suggest that the Yiluo River in northern China will have reduced annual runoff but with a wetter flood season under both 1.5 ℃ and 2.0 ℃ warming, while the Beijiang River in southern China will have a slight increase in annual runoff with a drier flood
season (Liu et al., 2017). The simulated runoff changes of the Yangtze River decrease under 1.5 ℃ warming; however, it shows opposite changes under 2.0 ℃ global warming (Chen et al., 2017).

The objectives involved in this paper address the following: (1) to detect the level of warming and the change in precipitation in four river basins with differing hydroclimatic characteristics under limiting global warming of 1.5 ℃ and 2.0 ℃, (2) to simulate the changes in river runoff under 1.5 ℃ and 2.0 ℃ warming among the four basins, (3) to estimate the uncertainty
constrained by global circulation models (GCMs) and RCPs, and (4) to quantify the difference in projected climate changes and simulated changes of river runoff in relation to 0.5 ℃ global warming difference among the four basins. To achieve these objectives, firstly, we analyze the projected changes in mean annual temperature and precipitation in the selected four basins under 1.5 ℃ and 2.0 ℃ warming. Secondly, we investigate the changes in simulated annual and monthly river runoff in the four river basins based on validated Soil Water Assessment Tool (SWAT). Finally, we quantify the uncertainties in
climate change projection and impacts on river runoff based on five GCMs under four RCPs.

## 2 Study basins and available data

### 2.1 Basins

Four basins that span a wide hydroclimatic gradient from dry to wet were selected as case studies in this research. The locations as well as the physical and hydroclimatic characteristics (based on the observation during 1961-2000) of the
selected basins are presented in Fig. 1 and Table 1.

The Shiyang River (SYR) basin is one of three inland river basins in Northwest China. The basin is dominated by a continental temperate arid climate and variable topography. The SYR has eight tributaries that originate in the Qilian Mountains, the total drainage in mountain area of which ($1.1 \times 10^4$ km$^2$) was selected as the study area. River discharge is derived mainly from precipitation and snow melt water in summer and from groundwater in winter. Of the eight tributaries
in the SYR basin, five have decreasing trends in annual streamflow, mainly because of reduced precipitation (Ma et al., 2008). The basin has lost much of its natural vegetation and it has undergone gradual desertification due to limited water resources, inappropriate human activities, and the arid climate, which together pose considerable threat to sustainable agricultural development (Zhu and Li, 2014).

The Chaobai River (CBR) basin is located on the North China Plain and it is a tributary of the Haihe River. The basin is
dominated by a continental temperate monsoon climate. The CBR originates from the Yanshan Mountain via two tributaries: the Chaohe River and the Baihe River. The total area of the basin above the Xiahui and Zhangjiafen gauging stations (about $1.4 \times 10^4$ km$^2$) was selected as the study area. This watershed is the source of more than half the water supplied to Beijing. Its

runoff has declined considerably during 1956-2004 because of climate change, land use and land cover change, and increased water consumption (Xu et al., 2014; Yang and Tian, 2009).

The Huaihe River (HHR) basin is an extensive flat plain located in a transition zone between the climates of North and South China. The basin is dominated by a warm temperate monsoon semi-humid climate. The upper region of the HHR above the Wujiadu gauging station, which has a drainage area of about $12.1 \times 10^4$ km$^2$, was selected as the study area. Climate change has led to increased severe storms, decreased intense droughts in HHR basin (Zhang et al., 2015).

The Fujiang River (FJR) is the tributary of the Yangtze River and originates from Min Mountain located in Southwest China. The FJR basin is dominated by a humid subtropical climate. The area above the Xiaoheba gauging station, which has a drainage area of $2.9 \times 10^4$ km$^2$, was selected as the study area. Because of the high population density, intensive agricultural practices, and decreasing precipitation, the observed river discharge has a decreasing trend; however, high-intensity and long-duration precipitation in this area frequently results in floods and associated landslides (Gao et al., 2017).

## 2.2 Available data

The consistent spatial dataset, such as the digital elevation model of China generated from topographic map with 1:250,000 scale, the harmonized world soil database with 30 arcsecond resolution (FAO/IIASA/ISRIC/ISS-CAS/JRC, 2008), and the digital land use map of China with 1:500, 000 were used for the parameterization of SWAT.

The observed discharge data were provided by the local authorities based on the Water Year Books. Monthly discharge records for selected gauging stations in the four basins (listed in Table 2) for the period of 1961–2001 were used for SWAT evaluation. The daily climate dataset (WATCH Forcing Data: WFD) (Weedon et al., 2010) with the resolution of 0.5 degree covered the period of 1958-2001 was obtained from the Water and Global Change Program. The WFD was used for driving SWAT hydrological model for the historical period, and also was used for the basis for GCMs output downscaling. Gridded reanalysis climate datasets have been use for hydrological modeling widely, and the WFD is considered an acceptable dataset for forcing hydrological models in comparison with gridded observation database (Essou et al., 2016). Furthermore, WFD has been widely used in climate change impact assessment at regional or catchment scale in China (Hao et al., 2018; Liu et al., 2017; Chen et al, 2017; Su et al., 2017). The comparison of mean annual and monthly temperature and precipitation based on WFD and meteorological observations (OBS) in the four river basins showed in Table S1 and Fig. S1. In this study, observations for 50 representative meteorological stations in the four river basins covered the period 1958-2017 were derived from the National Meteorological Information Centre of China of China Meteorological Administration. For the time period 1961-2001, WFD showed slight difference in the two river basins in Southern China, with about 1.3% and 2.1% lower in mean annual precipitation and 0.1 ℃ and 0.9℃ lower in mean annual temperature in the HHR and the FJR respectively. While in the two river basins in Northern China, there were less than 20% difference in mean annual precipitation (14.6% larger and 20% lower than observed meteorological observations), and 2.5℃ and 4.1℃ lower in mean

annual temperature in the SYR and the CBR. The monthly distribution showed general coherence in the seasonal pattern in temperature and precipitation between WFD and meteorological observation.

GCMs outputs were derived from the Inter-Sectoral Impact Model Intercomparison Project for five GCMs (HadGEM2-ES, IPSL-CM5A-LR, MIROC-ESM-CHEM, GFDL-ESM2M and NorESM1-M) under four RCPs (RCP2.6, RCP4.5, RCP6.0 and RCP8.5) (Warszawski et al., 2014). These models were selected to span global mean temperature change and relative precipitation change as effectively as possible (Warszawski et al. 2014). The FRC index (Fractional range coverage) of the five GCMs in ISI-MIP project is 0.75 and 0.59, respectively, which is better than the five GCMs randomly selected from CMIP5, and can reasonably represent the changes of regional average temperature and precipitation (McSweeny and Jones, 2016). These climate model outputs are spatially interpolated into 0.5 ° resolution and corrected using trend-preserving bias correction approach based on WFD for historical simulation (period 1950–2005) and for future projection (period 2006-2099) (Hempel et al., 2013). The downscaling climate data from GCMs showed very good coherence with WFD for the historical period 1961-2001 in the four river basins in this study (Table S2 and Fig. S2). There were slight differences in the WFD and downscaling climate data from GCMs for annual mean, maximum and minimum temperature in the four river basins, with less than 0.1℃ difference in the SYR, CBR and HHR, and 0.3℃ larger in the FJR. All five GCMs' historical downscaling data showed good agreement in temperature compared with WFD. For the annual precipitation, there was general wetter condition based on the five GCMs' historical downscaling data, with the magnitude less than 15%. The five GCMs' historical downscaling data could reproduce the monthly distribution of temperature and precipitation well. Such a subset provide climate information that can improve the understanding of both the total uncertainty of future climate impacts and the uncertainty constrained by the use of different GCMs and RCPs.

## 3 Methodology

### 3.1 Application of SWAT

The SWAT is a process-based semi-distributed hydrological model, which can simulate the river flow, water balance and nutrient transport at basin scale (Gassman et al., 2007). As an open and free tool, the SWAT is applied worldwide under various climatic conditions and hydrologic regime (Arnold et al., 2012).

The simulations using the SWAT model were forced by WFD climate data at a daily time step, and they were warm-up for the period 1958–1960. The SWAT models were then calibrated for the 1961–1990 and validated for 1991–2001 using monthly river runoff data from the gauging stations of the four basins. Forcing SWAT by WFD mainly based on the consideration of reducing the uncertainties of hydrological model parameterization caused by inconsistent climate forcing. Because climate model output was corrected based on WFD in the frame of ISI-MIP, and was used to force the calibrated SWAT model in the hydrological scenarios modeling. Forcing hydrological model with gridded climate/reanalysis climate data, and observed climate data result in different parameterization (Xu et al., 2010b), and has limited impact on performance of runoff simulation (Liu et al., 2012a; Liu et al., 2018, Wang et al., 2018).

Using sensitivity analysis procedures embed in SWAT resulted in the six most sensitive parameters (Table S3) in the hydrological model for each of the four rivers. There were two consistent sensitive parameters "CN2" and "GWQMN" among all four river basins which control the runoff process and soil water moving process respectively. However, there was consistent sensitive parameter for the two river basins located in northern China and southern China respectively, such as in the two river basins located in northern China, the common sensitive parameter was "ALPHA_BF" which reflect the groundwater flow response to changes in recharge. There were specific sensitive parameters for each river basin, such as the temperature related parameters for snow "SMTMP" and "TIMP" in the SYR. The definition of parameters showed in Table S4. The SWAT hydrological model were calibrated based on SWAT-CUP (SWAT Calibration and Uncertainty Programs) (Abbaspour., et al, 2007) to improve the fit between simulated and observed discharge. For the SYR, the observed monthly streamflow at the Jiutiaoling gauging station for the Xiyinghe tributary was used for model calibration and validation, while the parameterization was used for the entire the SYR. For the CBR, the observed monthly streamflow at the Xiahui gauging station for the Chaohe River and at the Zhangjiafen gauging station for the Baihe River were available for hydrological model calibration and validation separately (Hao et al., 2018). For the HHR and the FJR, the observed monthly discharge in the main stream at gauging station Wujiangdu and Xiaoheba respectively was used for model calibration and validation. However, the auto-calibration didn't result in satisfactory performance of the hydrological model in the SYR and the CBR. More extensive manual calibration was undertaken by manually varying the six most sensitive parameters in the SWAT which resulted in improvement in model performance, and a relative satisfactory fit between observed and simulated monthly river flow was obtained in the SYR and the CBR.

The coefficient of determination ($R^2$), Nash–Sutcliffe efficiency ($E_{ns}$) were used to measure the goodness-of-fit, and percentage of bias ($P_{bias}$) was used to assess systematic over- or under estimation and when the absolute value is applied it shows the magnitude (Green and van Griensven, 2008). In general, the model simulation is considered acceptable when the $E_{ns}$ values are greater than 0.5, $R^2$ should exceed 0.6, and the $P_{bias}$ less than ±20% (Moriasi et al., 2007). Furthermore, the performance of discharge simulation of SWAT was also compared by the graphical plots including monthly time series which reflects the month to-month sequencing, and flow duration curve which shows the frequency distributions of discharge.

Model performance statistics over the calibration and validation periods were all found "satisfactory" for the four basins (Table. 2). The performance statistics $E_{ns}$ and $R^2$ were both > 0.8 and considered highly acceptable for the two basins in southern China (i.e., the HHR and the FJR) for both the calibration and the validation periods. The same performance statistics were considered reasonably acceptable for the two basins in northern China (i.e., the SYR and the CBR) with efficiencies in the range 0.58~0.82. The $P_{bias}$ was generally less than 20% (excepted for the Baihe River for the calibration period) in the four rivers. The monthly time series for discharge during the calibration and validation period (Fig. S3) showed apparently well month to-month sequencing in the four rivers with general underestimation in monthly discharge in dry season in the two rivers located in northern China, and underestimation for flooding season in occasion years in the CBR. This was also reflected in the flow duration curve (Fig. S4), with large underestimation for the medium/lower and very high

flow for the CBR. Oppositely, there was overestimation in medium/lower flow in both the two river located in the Southern China, however, underestimate in higher flow in FJR.

It can be summarized that SWAT appears to capture successfully the underlying hydrology of the four river basins evaluated by the three statistic metrics and compared by the monthly discharge series, and flow duration curve. The successful application of the SWAT in different climate regions is considered adequate verification of the suitability of the model for future climate change impact on runoff in the four selected basins.

## 3.2 Climate change projection and runoff simulation

The future scenarios for limiting global warming of 1.5 ℃ and 2.0 ℃ were derived based on 30-year running mean of global mean temperature followed the methodology of Liu et al. (2017) for each one of the 20 combinations under four RCPs and five GCMs of the climate projection subset. Tab. S5 showed the averaged middle year of the 30-year samples for all GCMs under each RCPs of 1.5 ℃ and 2.0 ℃ global warming. There were 18 scenarios under 1.5 ℃ above preindustrial levels and 16 scenarios under 2.0 ℃. These scenarios were used to quantify the difference in the changes of the projected annual temperature and precipitation in the four river basins by comparing with the baseline period (1976-2005).

To indicate the overall magnitude and difference of the climate change projection under limiting global warming of 1.5 ℃ and 2.0 ℃, the projected changes in mean annual temperature and annual precipitation were quantified by the value of ensemble mean under all climate scenarios (Ave.), and the projected changes in maximum and minimum annual temperature and annual precipitation (Max. and Min.) among all climate scenarios. The uncertainty caused by RCPs was estimating using standard deviation of the mean of all GCMs under 1.5℃ and 2.0℃ global warming respectively, and the uncertainty constrained by GCMs was estimated using standard deviations of all RCPs under the two global warming, whereas the all source of uncertainty of climate change scenarios was estimating using the standard deviation of all the 18 and 16 climate scenarios under 1.5℃ and 2.0℃ global warming.

The hydrological simulation adopted the climate projection subset for the downscaling climate data, and the future climate scenarios from five GCM and validated SWAT models in the four basins, and projected the impact of climate change on river discharges. Generally, the hydrological simulations based on downscaling climate data from five GCMs for baseline period compared well with those based on WFD, and were acceptable subsequent hydrological projection (Tab. S6 and Fig. S5).The changes in averages of the annual and monthly runoff were compared based on the simulated runoff under all climate scenarios and with the simulated runoff based on the baseline period (1976-2005) from the five GCMs rather than the actual observed discharge data or simulated discharge forcing by WFD.

The simulated changes in mean annual runoff were quantified by the value of ensemble mean annual runoff of all climate scenarios under 1.5℃ and 2.0℃ global warming, and mean annual runoff under RCP 2.6, RCP4.5, RCP6.0 and RCP8.5 respectively, and mean annual runoff under GCM GFDL-ESM2M, HaDGem2, IPSL_CM5A-LR, MIROC-ESM-CHEM, and NorESM1-M respectively. The simulated changes in monthly runoff were analysis by the proportion of monthly runoff in

annual runoff using the mean of baseline period for 5 GCMs, and ensemble mean, maximum and minimum of simulated monthly runoff under all combined climate scenarios of GCMS and RCPs for 1.5℃ and 2.0℃ global warming, respectively.

## 4 Results

### 4.1 Projected climate change

The statistics of the projected climate change and uncertainties for the four basins from the 18 scenarios under 1.5℃ warming and the 16 scenarios under 2.0 ℃ warming were shown in Table. 3.

The results show substantial warming for all four basins under two thresholds global warming. The projected changes in ensemble mean annual temperature show 1.5℃ increase under 1.5℃ global warming and 2.2℃ increase under 2.0 ℃ warming for the SYR and the CBR. While, the projected changes in ensemble mean annual precipitation show 3% and 5%

increase under 1.5 ℃ warming, and 5% and 8% increase under 2.0 ℃ warming for the SYR and the CBR, respectively. The projected changes in ensemble mean annual temperature show 1.1 ℃ and 1.2 ℃ increase under 1.5 ℃ warming, and 1.8 ℃ increase under 2.0 ℃ warming for the HHR and the FJR. The projected changes in ensemble mean annual precipitation are minor for the HHR and FJR (i.e., <±3%). All statistics for the two basins in northern China indicate generally warmer and wetter conditions in future compared with the 'present day.' The two basins in southern China are projected to have less

warming and no consistent change in the projected ensemble mean annual precipitation.

The greatest range in projected changes in annual mean temperature occurs in the HHR, with the warming range of 0.3~1.6 ℃ under 1.5℃ warming and that of 0.7~2.3 ℃ under 2.0 ℃ warming among all projection scenarios. The projected range in annual temperature is also large for the SYR, with change in the range of warming 0.9~2.4 ℃ under 1.5 ℃ warming and that of 1.7~2.9 ℃ under 2.0℃ warming, respectively. There is no consistency in the direction of range

in projected annual precipitation change among the four river basins, with increases ranged 10% to 20% and decreases ranged −6% to −11%. For the two river basins in southern China, the range in projected change in annual precipitation is less than for the two basins in northern China.

The uncertainty is substantial in annual precipitation projection compared with that associated with annual temperature projection, with considerable dispersion among the scenarios. Comparing the uncertainty under limiting global warming

under 1.5 ℃ and 2.0 ℃, the former has larger uncertainties for the projected change in annual precipitation than that under the later; however, it is the opposite for the projected change in annual temperature.

There is generally larger uncertainty constrained by the GCMs (i.e., about 1~3 times) than associated with the RCPs for the projected annual precipitation for all four river basins. However, the uncertainty in annual temperature projection associated with the RCPs is larger in the SYR (about 2 times) and in the HHR (about 1.5~3.0 times) than constrained with the GCMs.

All these findings show the uncertainty in the projection of annual precipitation mainly constrained by GCM structure across the four river basins, whereas the dominance of the uncertainty associated with either the GCMs or the RCPs in the projection of annual mean temperature is dependent on the basin.

## 4.2 Simulated annual river runoff

Figure 2 shows the simulated ensemble mean annual river runoff based on all combined climate scenarios, and the average simulated annual river runoff of the four RCPs and the average of the five GCMs. The simulated ensemble mean annual runoff decreases for the SYR by about 25% and 33% under 1.5 ℃ and 2.0 ℃ warming, respectively, and the simulated change for the FJR shows a decrease of about 4% under 1.5 ℃ warming. The simulated ensemble mean annual river runoff shows an increase with magnitude of about 8% and 12% for the CBR and about 8% and 7% for the HHR under 1.5 ℃ and 2.0 ℃ warming, respectively.

The decrease in the simulated annual river runoff for the SYR occurs across all the combined scenarios, ranged from 0% to −72% under 1.5 ℃ warming and from −11% to −63% under 2.0 ℃ warming. For the other three river basins, the change in simulated annual river runoff ranges from an increase of 57% to a decrease of 34%. The smallest range occurs in the FJR, with a change in simulated annual river runoff in the ranged 10% to −17% and 11% to −11% under 1.5 ℃ and 2.0 ℃ warming, respectively. The largest range occurs in the HHR, with a change in simulated annual river runoff in the ranged from 57% to −34% under 1.5 ℃ warming and from 38% to −32% under 2.0 ℃ warming. The simulated change in annual river runoff in the CBR is in the ranged from 37% to −34% under 1.5 ℃ warming and from 39% to −20% under 2 ℃ warming.

The simulated change in annual river runoff for the mean of the four RCPs and the five GCMs shows consistent decrease in the range −61% to −14% under 1.5 ℃ warming and −56 to −18% under 2.0 °C warming for the SYR, with the largest decrease occurring under RCP2.6. The simulated annual river runoff under the mean of the four RCPs for the CBR shows consistent increase in the range 3% to 13% under 1.5 ℃ warming and 6% to 19% under 2.0 ℃ warming. For the HHR, the simulated annual river runoff under RCP2.6 shows reduction of −33% and −25% under 1.5 ℃ and 2.0 ℃ warming, respectively, whereas it increases under the other scenarios by 6% to 20% and 10% to 17%, respectively. For the FJR, the simulated annual river runoff shows reduction for all RCPs under 1.5 ℃ warming, but an increase for RCP4.5 and RCP6.0 under 2.0 ℃ warming.

The simulated annual river runoff for the CBR under HaDGem2 for the mean of the four RCPs shows decrease of about −9% and −2% under 1.5 ℃ and 2.0 ℃ warming, respectively, while that of the HHR under NorESM shows decrease of about −12%. However, for the FJR, most GCMs show reduction for the simulated annual river runoff in the ranged from 0% to −14% under 1.5 ℃ warming and from 0% to −5% under 2.0 °C warming, while any increase is no larger than 3%.

There is less uncertainty in the simulated annual river runoff among all the scenarios under 2.0 ℃ than that of 1.5 ℃ warming when quantified by standard derivation. The uncertainties associated with the RCPs are 1.3~2.6 times with those constrained by the GCMs for the SYR and the FJR, while for the CBR, the uncertainties constrained by the GCMs are 2~3 times those associated with the RCPs. For the HHR, the uncertainties associated with the RCPs are the largest under 1.5 ℃ warming, whereas those constrained with the GCMs are the largest under 2.0 ℃ warming.

## 4.3 Simulated seasonal river runoff

Figure 3 shows the change in the proportion (mean monthly percentage of annual runoff) of maximum, average, and minimum simulated river runoff based on all combined scenarios. For the SYR and FJR, the proportion shows no substantial change (i.e., <1.0%). For the CBR, a decrease occurs during May–July with magnitude of about 1.0% to 2.0%, and an increase occurs mainly in September and October with magnitude of <2.0% under 1.5 ℃ warming. Similarly, a decrease occurs during May–August with magnitude of 0.4% to 2.3% and an increase occurs in September with magnitude of about 2.0% under 2.0 ℃ warming. While, a decrease occurs mainly during June–August for the HHR, with magnitude of about 1.0% to 3.5% and 1.2% to 3.4%, while an increase occurs in May with magnitude of about 2.0% and in September with magnitude of <5% under 1.5 ℃ and 2.0 ℃ warming, respectively.

For all months, there are generally larger ranges for the mean monthly percentage of annual runoff for 1.5 ℃ warming. These results indicate the uncertainties in simulated monthly runoff are larger under 1.5 ℃ warming than under 2.0 ℃ warming.

## 5 Discussion

### 5.1 Climate change impact on runoff

Chen et al. (2014) analyzed the effects of climate change on runoff in the Asian monsoon region. They indicated that different basins respond differently to the same climate change scenario. For example, they found that the change in runoff of the Haihe River basin in northern China is highly sensitive to precipitation and temperature. It was established that a considerable increase in precipitation (about 4%) would be required to keep runoff unchanged in this semi-humid basin in Northeast China, while a smaller precipitation increase (about 2.8%) would be required to maintain runoff in wetter basins in South China. Precipitation is the main input of surface water resources and evapotranspiration (ET) is the main output. Previous studies have explored the climatic impacts of ET and runoff in China. For example, Liu et al. (2012) analyzed the environmental stress on ET and runoff over eastern China for 1961–2005. They found ET increased in most river basins, while runoff increased in the Pearl River and the southeast river basins in southern China but it decreased in the basins of the Haihe and Huaihe rivers in northern China. It was determined that climate change was the dominant factor governing the long-term trend of ET and runoff in southern China. Ma et al. (2008) indicated that decreased precipitation and increased potential ET contribute most to the observed reduction of streamflow in SYR in northwest China.

The four river basins in this study represent climate from dry to wet, and the response of runoff to precipitation change is also consistent with the previous findings (Chen et al., 2014; Liu et al., 2012; Ma et al., 2008) that more increase in precipitation need to maintain runoff in drier basins. In this study, a smaller precipitation change (±3%) would maintain a change in runoff of about 7% and 8% in the HHR and of about 0% and −4% in the FJR under 1.5 ℃ and 2.0 ℃ warming in the wetter area; while, for the CBR in a semi-humid climate area, an increase in precipitation of about 5% and 7% would

maintain an increase in runoff of about 8% and 12% under 1.5 ℃ and 2.0 ℃ warming. Moreover, for SYR in the arid climate region, an increase in precipitation of about 5% and 7% companied with a decrease in runoff of about −33% and −25% in the SYR.  Further analysis of ET simulation (Fig.4) indicated a general increase in simulated ET in all four basins. However, the magnitude of the simulated change of ET varies across the basins, i.e., it is larger in the two basins in north

China than in the two basins in south China. For the two rivers located in northern China, the simulated change of ET in the SYR shows increase of 21% and 13%, while that of the CBR shows increase of 4% and 6% under 1.5 ℃ and 2.0 ℃ warming, respectively, which implies the increase in simulated ET contributes most to the decrease in simulated annual runoff in the SYR.

### 5.2 Uncertainties in the quantitative assessment

This study followed the top-down methodology that common used in IPCC AR4 and AR5 WGII report. Within the IPCC AR 4 and AR5 water sector, most hydrological projection studies use the precipitation and temperature downscaled from GCMs to driven hydrological models. This study adopted climate projection information derived from Inter-Sectorial Impact Model Intercomparison Project (ISIMIP). Climate outputs are spatially interpolated into 0.5 °×0.5 ° resolution and corrected using trend-preserving bias correction approach based on reanalysis dataset WFD. The WFD was also the climate forcing to

calibrate and validate of SWAT hydrological model. There were multi sources of uncertainties in climate change impact assessment in this study. Considering the challenge to address uncertainties for all sources, which we only focus on the uncertainties constrained by GCMs and RCPs. Certain uncertainty source were not investigated, such as the climate forcing, hydrological model structure and parameterization, GCM structure.

Climate forcing is one of major uncertainty in quantitative assessment of climate change impact (Müller Schmied et al.,

2014). The complex terrain in the four river basins makes it difficult for WFD to reach very satisfactory agreement with station based observation. The comparison of WFD with climate observation from meteorological station showed reasonable agreement (Fig. S1 and Table S1), but, there was both underestimation and overestimation in precipitation and temperature based on WFD. This could induce the uncertainty in the hydrological simulation, such as difference in the ET simulation in SYR (Fig. S5). Furthermore, the validated SWAT driven by downscaling climate data from GCMs for baseline period and climate scenarios under

1.5℃ and 2.0℃ global warming. Although, the method used for estimated the projected changes in runoff could avoid systematic errors that the SWAT model would introduce in comparing the projection period with the baseline period. However, uncertainty in runoff simulation would spread to the runoff assessment.

Meanwhile, the application SWAT in four river basins covered various climate and environmental condition may result in uncertainty constrained by hydrological model structure and parameterization. Li et al. (2016) indicated that frozen soil meltwater

accounted for about 20% of river runoff during the flood season, while glacier meltwater contributed about 3% in the SYR. There was a few cases showed that SWAT could be used in snowmelt-dominated streamflow (Wang and Melesse, 2005; Tolston and Shoemaker, 2007; Grusson et al., 2015), a few previous researches have indicate that SWAT model did not adequately predict winter flows or snowmelt-dominated runoff in several watershed (Peterson and Hamlett, 1998; Srivastava et al., 2006;

Chanasyk et al., 2003; Benaman et al., 2005) , which could be one reason that the low values of $E_{ns}$ for the SYR and the CBR in the northern China with cold and dry winter.  This also could induce the uncertainty in the river runoff simulation. Furthermore, the glacier meltwater process was not considered in SWAT-based simulations in this study, which would enlarge the uncertainty in runoff assessment.

Moreover, GCM selection would introduce uncertainty and influence the range of climate change impact assessment (Todd et al., 2011). The five GCMs used in this study, which captured 50 to 90% of the full range of future projections of 36 CMIP5 GCMs for temperature and 40 to 90% of the full range of future projections for precipitation in the four river basins (Fig. 1 in McSweeney and Jones, 2016). Furthermore, Liu et al. (2017) compared the changes of precipitation and temperature by five GCMs used in this study with those of other 19 CMIP5 GCMs. The results showed that the five GCMs

covered the range of GCMs from CIMP5 well for global mean precipitation and temperature during 2020-2050 for RCP2.6 and RCP4.5. The information indicates the importance for reducing uncertainty associated with the choice of an applied GCM. At the basin scale, prioritising or weighting GCMs may be considered on the basis of detailed analyses of the ability of an individual GCM to represent a specific characteristic of regional climate of interest (e.g., multi-annual or decadal variability).

**6 Conclusion**

The 2.0 ℃ warming scenario caused more substantial warming than the 1.5 ℃ warming scenario in all four studied basins. For the two basins located in northern China, the 0.5 ℃ global warming difference caused warming of 0.7 ℃ in the local ensemble mean temperature; however, in southern China, this difference caused warming of 0.6 ℃. The 0.5 ℃ global warming difference will cause consistently wetter conditions, with projected precipitation amounts about 2% greater for the

four basins, although the projected changes in annual precipitation are minor in southern China compared with the increases in northern China.

The 2.0 ℃ warming caused a decrease of 8% and 1% in the simulated ensemble mean annual runoff in the SYR and the HHR compared with 1.5 ℃ warming, while it caused 4% increasing in the CBR and the FJR. Climatic–hydrological interaction increases the complexity of changes in simulated annual runoff; however, the 0.5 ℃ global warming difference

will coincide with a "wet-get-wetter" and "dry-get-drier" response in the two basins in northern China, and it will moderate the simulated annual runoff in the two basins in southern China. There is no shift in seasonal runoff pattern attributable to the effects of projected changes in climate under 1.5 ℃  and 2.0 ℃ warming; however, the monthly runoff percentage does change in the CBR and the HHR in some months.

The range of projected annual temperature is largest for the HHR and the SYR, with the uncertainties dominated mainly by

the RCPs. Conversely, the ranges are smallest in the CBR and the FJR, with the uncertainties mainly constrained by the GCMs. Although, the range in the projected change in annual precipitation is smaller in the two basins in southern China than in the two basins in northern China, the GCMs constitute the major source of the uncertainties in the projection of

annual precipitation for the four river basins. Even under the limiting global warming thresholds of 1.5 ℃ and 2.0 ℃, the uncertainties in the projected annual temperature at local or regional scale are dominated by either the GCMs or the RCPs; however, the uncertainties in local and regional projected annual precipitation are mainly constrained by GCM structure. The 0.5 ℃ global warming difference will generally reduce the uncertainties in the projected change in annual precipitation.

5  There is less uncertainty in the simulated change in runoff among all scenarios under 2.0 ℃ warming compared with 1.5 ℃ warming. This is consistent with the uncertainty in the projected annual precipitation. However, the uncertainties, dominated by the GCMs for the Chaobai River and constrained by the RCPs for the SYR and the FJR, limit confidence in the projected annual runoff for the four studied river basins.

**Acknowledgments:** We wish to thank the Inter-Sectoral Impact Model Intercomparison Project (ISIMIP), which make the data of the five GCMs available. This study was jointly supported by the National Key R&D Program of China (2016YFE0102400), research project of the Meteorological Public Welfare Industry in China (GYHY201406021), and climatic change project of the China Meteorological Administration (CCSF201810).

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

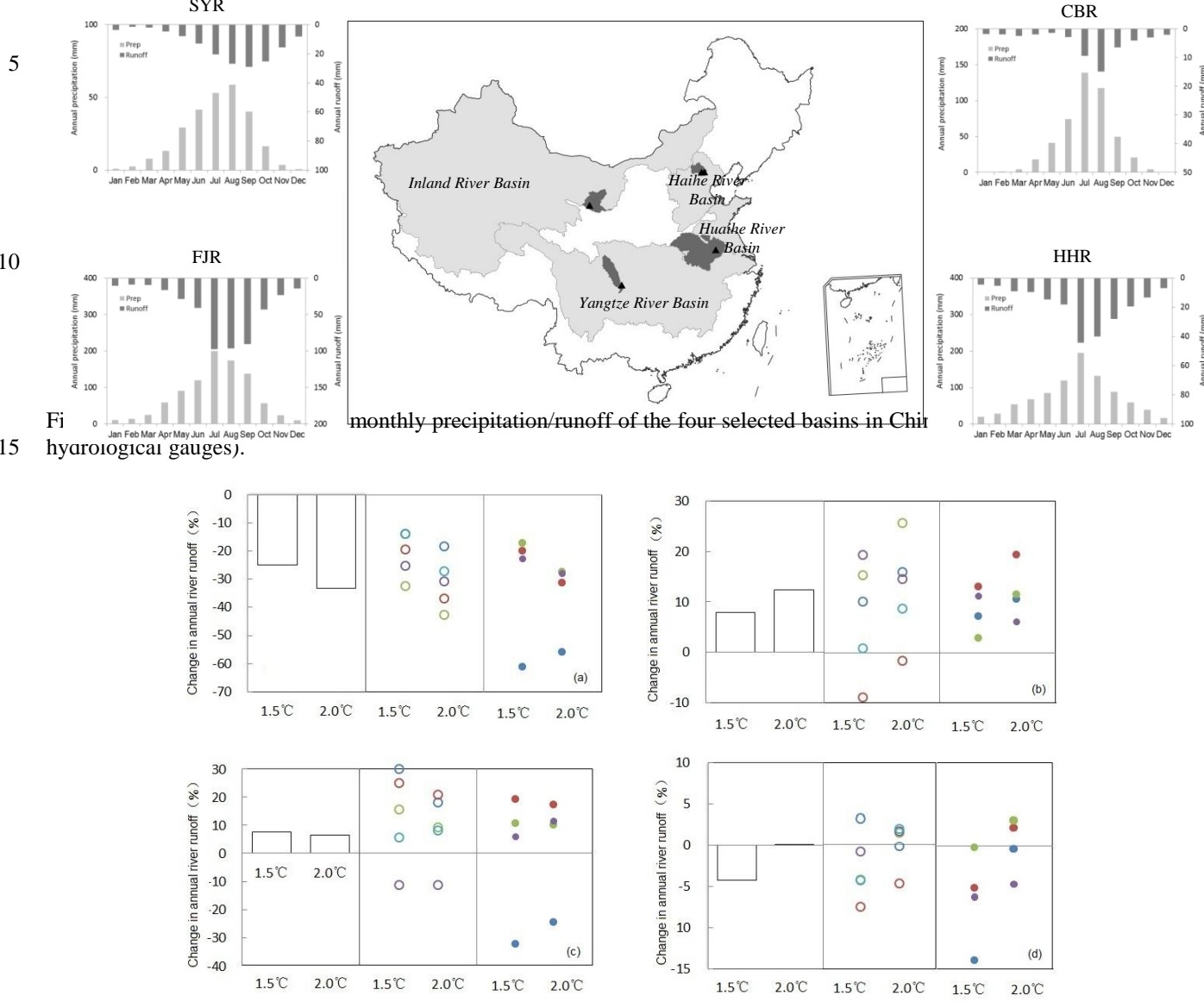

Fi                                   monthly precipitation/runoff of the four selected basins in Chir
15   hydrological gauges).

Figure 2. Changes in simulated annual river runoff: (a) SYR, (b) CBR, (c) HHR, and (d) FJR under 1.5 ℃ and 2 ℃ global warming. (Baseline: 1976–2005; columns represent the simulated changes in mean annual river runoff for all combined scenarios of GCMs and RCPs ; hollow circles colored dark blue, red, green, blue, and purple represent the changes in mean
20   annual runoff simulated by five GCMs: GFDL-ESM2M, HaDGem2, IPSL_CM5A-LR, MIROC-ESM-CHEM, and NorESM1-M, respectively; solid circles colored dark blue, red, green, and purple represent the changes in mean annual runoff simulated under four RCPs: RCP2.6, RCP4.5, RCP6.0, and RCP8.5, respectively).

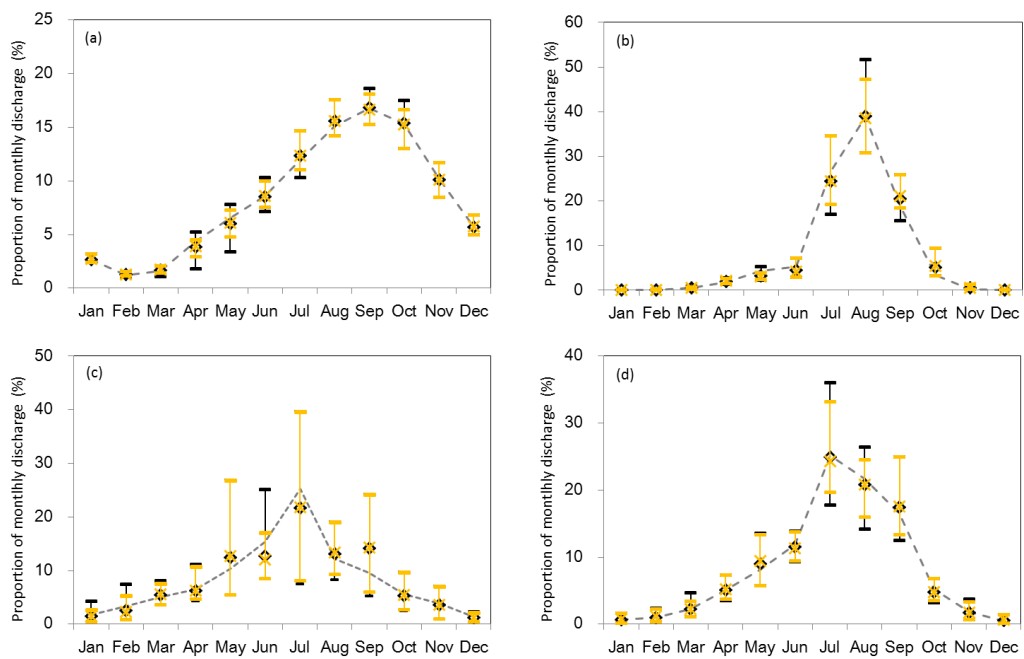

Figure 3. Simulated proportion of monthly river runoff in annual runoff: (a) SYR, (b) CBR, (c) HHR, and (d) FJR under 1.5 and 2.0 ℃ global warming. (Baseline: 1976–2005; dotted line: mean of baseline for 5 GCMs, bars colored black and yellow show the maximum and minimum values of all simulated monthly runoff for all combined climate change scenarios of GCMs and RCPs; black diamonds and yellow crosses represent the mean values for monthly runoff for all combined climate change scenarios of GCMs and RCPs) .

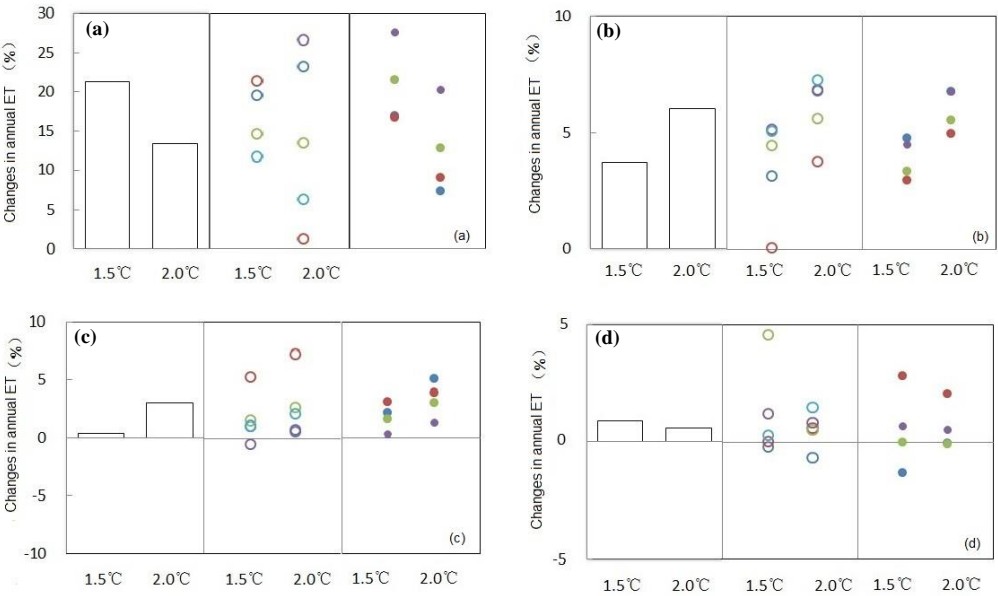

Figure 4. Same as in Figure 2 but for change in simulated annual ET.

Table 1. Hydroclimatic characteristics of the four selected basins.

| Basin | Total Area (Km$^2$) | Study Area (Km$^2$) | Altitude(m) | | | 1961-2000 Average(mm) | |
|---|---|---|---|---|---|---|---|
| | | | Max | Mean | Min | Precipitation | Runoff |
| SYR | 41,600 | 11,000 | 5090 | 2448 | 1398 | 498 | 180 |
| CBR | 19,354 | 13,846 | 2266 | 930 | 38 | 469 | 53 |
| HHR | 144,900 | 121,330 | 2099 | 106 | 11 | 910 | 203 |
| FJR | 36,400 | 29,488 | 5541 | 1027 | 242 | 964 | 481 |

Table 2. Goodness of fit of SWAT simulations for monthly runoff of the SYR, CBR, HHR, and FJR.

| Basin | Calibrated area | | | Calibration (1961-1990) | | | Validation (1991-2001) | | |
|---|---|---|---|---|---|---|---|---|---|
| | River | gauging | Area (km$^2$) | $R^2$ | $E_{ns}$ | $P_{bias}$ | $R^2$ | $E_{ns}$ | $P_{bias}$ |
| SYR | Xiyinge | Jiutiaoling | 1,077 | 0.65 | 0.82 | 1% | 0.71 | 0.58 | 7% |
| CBR | Chaohe | Xiahui | 5,340 | 0.63 | 0.63 | 1% | 0.68 | 0.65 | 8% |
| | Baihe | Zhangjiafeng | 8,506 | 0.60 | 0.56 | 25% | 0.77 | 0.61 | -2% |
| HHR | Huaihe | Wujiadu | 121,330 | 0.88 | 0.87 | 16% | 0.86 | 0.81 | 8% |
| FJR | Fujiang | Xiaoheba | 29,488 | 0.94 | 0.87 | 1% | 0.93 | 0.87 | 5% |

Table 3. Projected changes in annual mean temperature and annual precipitation for the four basins under 1.5 ℃ and 2.0 ℃ global warming.

| Basin | Global warming | Annual mean temperature | | | | | | Annual precipitation | | | | | |
|---|---|---|---|---|---|---|---|---|---|---|---|---|---|
| | | Changes (°C) | | | Uncertainty | | | Changes (%) | | | Uncertainty | | |
| | | Ave. | Max. | Min. | All | GCMs | RCPs | Ave. | Max. | Min. | All | GCMs | RCPs |
| SYR | 1.5°C | **1.5** | 2.4 | 0.9 | 0.36 | 0.16 | 0.38 | **3** | 18 | -11 | 7.0 | 6.6 | 5.0 |
| | 2.0°C | **2.2** | 2.9 | 1.7 | 0.32 | 0.13 | 0.29 | **5** | 15 | -6 | 6.0 | 4.7 | 2.1 |
| CBR | 1.5°C | **1.5** | 1.8 | 1.1 | 0.22 | 0.20 | 0.02 | **5** | 17 | -11 | 7.3 | 6.0 | 2.2 |
| | 2.0°C | **2.2** | 2.8 | 1.7 | 0.33 | 0.15 | 0.06 | **7** | 20 | -8 | 6.3 | 3.6 | 2.0 |
| HHR | 1.5°C | **1.1** | 1.6 | 0.3 | 0.35 | 0.21 | 0.30 | **0** | 13 | -9 | 6.3 | 4.4 | 4.3 |
| | 2.0°C | **1.8** | 2.3 | 0.7 | 0.38 | 0.12 | 0.35 | **3** | 13 | -9 | 6.3 | 3.7 | 3.7 |
| FJR | 1.5°C | **1.2** | 1.7 | 0.8 | 0.23 | 0.24 | 0.06 | **-2** | 12 | -10 | 5.6 | 5.0 | 3.8 |
| | 2.0°C | **1.8** | 2.2 | 1.3 | 0.28 | 0.17 | 0.10 | **0** | 10 | -6 | 4.6 | 4.1 | 2.1 |