# Peer review of "Assessment of climate change impact and difference on the river runoff in four basins in China under 1.5 $^{\circ}$ C and 2.0 $^{\circ}$ C global warming"

_Hydrology and Earth System Sciences, 2018_

## Referee Comment (RC1) · Anonymous Referee #1 · 28 Oct 2018

There are a few general comments:

1. Was the conformity assessment of meteorological characteristics (for example, air temperature, precipitation, and other input variables of the SWAT model) from the WFD dataset to the observed values on the meteorological monitoring network carried out?

2. Was the assessment of reliability of meteorological conditions reproduction according to the GCMs data for the baseline period 1976-2005 in comparison with the WFD dataset carried out? as well as the annual and seasonal water regime of the rivers according to the simulation results of the SWAT model? This can be extremely important for future calculations.

[Figure]

3. Probably, the low values of the Nash-Sutcliffe efficiency for gauges with a smaller catchment area (the Shiyang and Chaobai rivers) than for larger ones are explained by the insufficiently detailed grid of the meteorological data (0.5 degree).

4. How were the threshold values of 1.5 °C and 2 °C determined according to GCMs? at the end of the XXI century or during?

5. How different are the sets of calibrated parameters of the SWAT model for the four study rivers?

I believe that the authors' responses to these comments will allow us to more objectively evaluate the obtained simulation results and understand the features of the nonlinear response of hydrological systems to climate change, which will further increase the scientific level and significance of the results of this article.

---

## Referee Comment (RC2) · D. A. Post (Referee) · 11 Nov 2018

There are some fundamental problems with this paper that make me very uneasy about recommending it for publication. Firstly, the choice of five GCMs from a much large available set of AR5 projections needs to be justified. At the very least, we need to know why these five were chosen and whether they differ from the larger set in terms of their future projections. I'd also like to know how these five perform under historical conditions. Related to this, we need to know how the historical projections from the GCMs compare with the historical data used to calibrate the model. Section 3.2 is very unclear about this. If the historical GCM data is wildly different from the historical

calibration data, I cannot see how it can be used to assess current conditions and therefore used to assess projected changes. Similarly, the SWAT calibration statistics are rather poor. The biases in calibration of 16% and 25% are much greater than the projected changes in runoff. How can we have any confidence in these projected changes when the calibrations are unable to get even the correct volume of runoff? Using the model to project seasonal changes when historical seasonal statistics were not examined is also unacceptable. The inadequacy of the model for use in climate change studies is re-iterated by the -11 to +18% change in precipitation for the Shiyang River leading to reductions in annual flow of 10% to 60%. This is not credible, and clearly the model is giving too much weighting to the impact of increases in PET.

There is some value in the estimates of changes in temperature and precipitation across the four river basins, but the large bias in the hydrological model calibrations means that I cannot see how these changes in precipitation can be converted into changes in even annual runoff. Also, the changes in temperature and precipitation are predicated on just five GCMs, and we would need to know where these fall within the range of all GCMs in AR5.

The authors have pretty much ignored the very large body of work emanating from Australia, the US and Europe on estimating impacts of climate change on water availability. I'd strongly suggest they go back and read the approaches that have been used elsewhere and modify their approach based on this.

Specific comments: From the abstract, it appears as if the focus of the paper is on the impact of an additional 0.5 degrees global warming, not the impact of 1.5 and 2.0 degrees compared to current conditions. However, the paper does not focus on this 0.5 degree difference.

Line 6. The target of 1.5 degrees is thought to be the one which might limit dangerous climate change impacts, not 2 degrees as proposed here. In fact, the comparison of 2 to 1.5 degree warming can be considered to be 'what if' we don't manage to keep

to 1.5 degrees of warming? What might the additional 0.5 degrees do? That could be a useful focus of the paper, however the problems raised above mean that this cannot be done with the current approach.

Considering the enormous range of projected changes across the AR5 GCMs, the reader needs to know why the authors selected the five GCMs used in this study. Were they just more accessible? How does the range of projections from those GCMs compare to the larger set of GCMs in AR5? Without knowing this, we have no idea if these projected changes represent a wetter/drier hotter/cooler part of the spectrum of future climate change projections.

One of the key issues in hydrological modelling studies is whether the model is able to represent the current conditions well enough to be able to be used in climate change studies. In this paper, the authors claim that the model calibration and validation results are 'satisfactory'. While this may be true to some extent for the Huiaihe and Fujiang Rivers, the calibration and validation statistics for the other rivers are poor at best (remembering that they are only attempting to produce monthly, not daily streamflow). Even more concerning however in a study such as this one is that the calibration bias is 25% for the Baihe and 16% for the Huaihe River. As the projected change in annual runoff is much less than that, I cannot see how the authors can justify using such a poor calibration. I am not familiar with the WFD climate data, but I strongly suspect that is the main reason the calibrations are so poor. Are there any other datasets (local precipitation for example) that could be used instead? Also, was SWAT run on a monthly or daily basis? No information is provided. It is not at all clear which precipitation data were used to drive the SWAT model under the future climate scenarios. Section 3.2 is confusing and not at all clear. Did the authors simply take the precipitation from the climate models directly and run SWAT for both the historic and future scenarios? If so, how did these precipitation projections, particularly the historical 'projections' compare to those used in the historic calibration? If they were significantly different, this gives us some information about how well the GCM's are predicting historical conditions and

some confidence (or likely not) in their use in the future projections.

Figure 2 shows that all future projections for the Shiyang River are for reductions in annual flow (of between 10% and 60%), but Table 3 states that annual precipitation shows a range of changes from an 11% decrease to an 18% increase. If the modelling indicates that an 18% increase in precipitation will lead to a reduction in annual runoff then the model is clearly inadequate for use in climate change studies.

While it is written well overall, considering the authors all presumably have English as a second language, there are a few sentences that do not make sense, such as line 23-25.

Figure 1. What does the light grey shading signify? What is the inset attempting to show?

---

## Referee Comment (RC3) · Anonymous Referee #3 · 23 Nov 2018

General comment This study attempts to investigate climate change impact on river runoff in four river basins across China, using hydrological model simulations forced by meteorological data representing 1.5 and 2 C global warming based on 5 global climate models (GCMs) under 4 emission scenarios (RCPs). The objective is further to quantify the uncertainties in the projected changes given by the GCMs and RCPs.

There are a couple of general problems in the study that need to be addressed in order to be accepted: * There is very little information about how the hydrological model was calibrated. Which parameters were calibrated, and which criteria were used for the calibration? The inconsistent response in river runoff to the increasing precipitations

over the study basins suggests that the results are strongly controlled by changes in evapotranspiration (as a result of changes in temperature and water availability). Thus we need to know more about how evapotranspiration is simulated in the model, and if and how parameters related to evapotranspiration were part of the calibration. There is also a lack of evaluation of how well the model manages to explain the observed changes in river runoff, which are referred to in the introduction. As a summary, it is doubtful if the current model is adequate for the impact study presented in the paper.

* A related problem is the selection of meteorological forcing data used in the study. First of all, there is no assessment presented of the agreement during the historical period between the data used for the model calibration (WFD) and the data used for the climate projections - thus we cannot assess to what extent the calibrated model us suitable for assessing the climate change impact with these data. Secondly, there is very little motivation or details given regarding the selection of the GCM models, or the selection of the 30 year periods representing 1.5 and 2.0 C warming, respectively. The selection of GCM models should be crucial for the quantification of uncertainties, which is pointed out as one of the objectives of the paper.

* In addition to the methodological and presentation issues, the paper is very uneven in the quality of the English writing, which makes it difficult to understand some of the statements.

Specific comments

Figure 1: I would assume that the dark grey areas represent the study basins, but what is represented by the light grey area? I would further assume that the basin locations following the position of the surrounding graphs, but I cannot be sure without consulting the text. What is presented in the small embedded graph? It looks like some mistake.

Methodology section: * How was the model calibrated? Which model parameters? Which objective function was used in the calibration? * Please give some more explanation how the 30 year periods were selected for the different global warming thresholds - as well as how the standard deviations referring to the GCMs and the RCPs, separately were quantified. How was the standard deviation originating from the GCMs and the RCPs aggregated into the values presented in Table3?

Results section: I would prefer not to use sentences that only refer to a table or a figure without describing any of the results. Describe the result in the text and use the tables and figures as support. For instance, I would recommend to refer more directly to the specific results in Table 3 that supports the various statements in section 4.1.

---

## Author Comment (AC1) · 28 Dec 2018

Anonymous Referee #1: We appreciate the Referee #1's comments and suggestions on our manuscript. The following are our point-point replies, with reference to the order of the comments by the reviewer. Comment 1: Was the conformity assessment of meteorological characteristics (for example, air temperature, precipitation, and other input variables of the SWAT model) from the WFD dataset to the observed values on the meteorological monitoring network carried out? Response: Thanks for this suggestion. Gridded reanalysis climate datasets have been use for hydrological modeling widely. WFD are derived from the ERA-40 reanalysis product via sequential interpolation

to half-degree resolution, elevation correction and monthly-scale adjustments based on CRU (corrected-temperature) and GPCC (precipitation) monthly observations. WFD has been compared with CRU data and GPCC data for monthly temperature, wet days, and precipitation totals etc. WFD is considered an acceptable dataset for forcing hydrological models in comparison with gridded observation database at global scale (Essou et al., 2016). Furthermore, WFD has been used in climate change impact assessment at regional or catchment scale in China (Hao. et al., 2018; Liu et al., 2017; Chen et al, 2017; Su et al., 2017). Table 1S1 and Figure S1 showed the comparison of mean annual and monthly temperature and precipitation based on WFD and meteorological observations. The annual mean temperature and annual precipitation from the WFD forcing data were 2.5°C lower and 14.5% higher, respectively, than those observed in the Shiyang River in 961-2001; those were 4.1°C and 20% lower in the Chaobai River, respectively, and those were similar in the Huaihe and Fujiang Rivers. The distribution of monthly mean temperature and monthly precipitation showed lower values for the Shiyang and Chaibei rivers for each month, while showed good agreement in the Huaihe and Fujiang river. Previous research indicates that the gridded climate dataset can be used in hydrological modeling, and the performance of hydrological model will improve by model calibration and validation (Xu et al., 2011). Comment 2: Was the assessment of reliability of meteorological conditions reproduction according to the GCMs data for the baseline period 1976-2005 in comparison with the WFD dataset carried out? as well as the annual and seasonal water regime of the rivers according to the simulation results of the SWAT model? This can be extremely important for future calculations. 1) Considering the WFD covers the period of 1958-2001, we compared the downscaled climate simulation from 5 GCMs with WFD during 1961-2001. Table S2 and Figure S2 showed the agreement of WFD with the historical simulation of 5 GCMs at mean annual scale and monthly scale. The downscaled GCMs data showed very good agreement with WFD for both the mean annual temperature and precipitation. The differences in mean annual temperature between WFD and downscaled 5 GCMs output were -0.03°C∼0.36°C for the

four river basins, while those of mean annual maximum and minimum temperature were -0.02°C∼0.29°C and -0.07°C∼0.41°C respectively. There were general overestimates for mean annual precipitation based on the downscaled historical climate simulation from 5 GCMs. The difference in mean annual precipitation were 5.2%∼14.8% between WFD and downscaled historical climate simulation from 5 GCMs in the Shiyang River, those were 6.3%∼9.7% in the Chaobai River, 3.9%∼5.4% in the Huaihe River, and 5.6%∼11.0% in the Fujiang River. The downscaled GCMs historical climate simulation fitted the distribution of mean monthly temperature with WFD very well during the 1961-2001. The most of month with precipitation were overestimated by the downscaled GCMs simulation than underestimated for the four river basins, especially for the precipitation in spring and autumn. However, those differences in monthly precipitation based on WFD and downscaling climate historical simulation from five GCMs didn't change the seasonal pattern of precipitation. The downscaled GCMs output from ISI-MIP were reliable unified set of climate drivers to allow a consistent analysis of climate change impacts on water resource at basin scale. 2) For subsequent hydrological simulation, this study adopted downscaled GCMs data derived from five GCMs and validated SWAT models, and projected the impact of climate change on river discharges. The detailed comparison in simulated annual runoff and evapotranspiration (ET) based on WFD and downscaling climate data from five GCMs showed in Table S6. The results indicated the difference in simulated runoff were large in the Shiyang River for all five GCMs and for two GCMs in the Chaobai River and one GCM in the Fujiang and Huaihe River. This maybe caused the relative poor performance of SWAT calibration and validation in the two northern river basins and contributed by the uncertainties in GCMs data downscaling. Runoff simulated based on GCM HadGem2-ES showed big difference with those of WFD for all the four river basin, which make the subsequent hydrological projection with big uncertainty. The simulated ET based on WFD and downscaling climate from five GCMs showed the similar conditions, however, there were generally underestimated ET for the overestimated runoff to keep the water balance. The monthly distribution of

ET were not changed for the most simulated runoff based on WFD and downscaling GCMs climate data. The simulated ET was underestimated for the Shiyang River, especially during the summer, with the peak of ET earlier based on the simulation of GCM HadGem2-ES. The simulated monthly ET based on GCH MIROC-ESM_CHEM also showed earlier peak in the Fujiang River. Generally, the hydrological simulations based on downscaling climate data from five GCMs for baseline period compared well with those based on WFD, and were acceptable subsequent hydrological projection. The annual and monthly runoff changes were calculated using 30 years of projected monthly runoff over each simulation under all climate scenarios of five GCMs and four RCPs, and then compared with the discharge simulated based on downscaling climate data derived from five GCMs for baseline period rather than the actual observed discharge data or simulated discharge data based on WFD. This technique was used to avoid systematic errors that the SWAT model would introduce in comparing the projection period with the baseline period. We have supplemented the comparison of WFD and downscaling climate data from five GCMs, and the further runoff and ET simulation based on these dataset in the revised manuscript. Comment 3: Probably, the low values of the Nash-Sutcliffe efficiency for gauges with a smaller catchment area (the Shiyang and Chaobai rivers) than for larger ones are explained by the insufficiently detailed grid of the meteorological data (0.5 degree). ResponseïijŽI really appreciate this comment. 3) I agreed with the reviewer's comment that high resolution climate forcing maybe improve the hydrological model performance. To avoid the uncertainty caused by the inhomogeneity of the spatial distribution of meteorological stations, the gridded climate dataset with 0.5 degree resolution was used to force the SWAT hydrological model in this study. I prefer to do further investigate using available high resolution climate forcing to calibrate and validate SWAT hydrological model the four river basins to compare the hydrological model performance with forcing with 0.5 degree WFD dataset in the future research. 4) Furthermore, whether the hydrological model structure can reflect the specific hydrological process is the key factor to deter-mine the hydrological model performance. There was a few cases showed that SWAT

could be used in snowmelt-dominated streamflow (Wang and Melesse, 2005; Tolston and Shoemaker, 2007; Grusson et al., 2015), a few previous researches have indicate that SWAT model did not adequately predict winter flows or snowmelt-dominated runoff in several watershed (Peterson and Hamlett, 1998; Srivastava et al., 2006; Chanasyk et al., 2003; Benaman et al., 2005) , which could be one reason that the low values of the Nash-Sutcliffe efficiency for the Shiyang and Chaobai rivers in the northern China with cold winter. We have explained the reason for the low values of the Nash-Sutcliffe efficiency for Shiyang and Chaibai rivers in the revised manuscript. Comment 4:How were the threshold values of 1.5°C and 2°C determined according to GCMs? at the end of the XXI century or during? Response: Future time horizon of global warming of 1.5°C and 2°C is derived based on 30-year running mean of global mean temperature (GMT) for each one of the 20 combinations of four RCPs (RCP2.6, RCP4.5, RC6.0 and RCP8.5) and five GCMs. When the GMT anomaly of 30-year running mean relative to pre-industrial level reaches the threshold of 1.5°C or 2°C, the 30-year window is sampled as corresponding time horizon of global warming scenario. Then year in Table S5 is estimated by averaging all center-years of the 30-year samples for all GCMs under each RCP and under each global warming scenario. Among these 20 combinations, 16 scenarios show mean GMT increases exceeding the threshold of 2°C above pre-industrial level, and 18 scenarios exceed the threshold of 1.5°C. But the changes in projected variables (annual temperature and precipitation) are quantified relative to present day (1976 – 2005) instead of pre-industrial period in this research. We have supplemented the methodology about define the 1.5°C and 2.0°C warming in the revised manuscript. Comments 5: How different are the sets of calibrated parameters of the SWAT model for the four study rivers? ResponseïijŽUsing sensitivity analysis procedures embed in SWAT resulted in the six most sensitive parameters (Table S1) in the hydrological model for each of the four rivers and then used for model calibration. The consistent sensitive parameters among all four river basins included parameter "CN2" and "GWQMN" which control the runoff process and soil water moving process respectively. The consistent sensitive parameters

for the two river basins located in the northern China was parameter "ALPHA_BF" which reflect the groundwater flow response to changes in recharge; for the two river basins located in southern China, the common sensitive parameter was "RCHRG_DP" which was a coefficient that define the aquifer percolation fraction. However, because the differences in meteorological and hydrological conditions, topography and soil properties, there was specific sensitive parameters for each river basin, such as for the Shiyang River, the specific sensitive parameters were "SMTMP" and "TIMP" which are temperature related parameters for snow; for the Chaobai River, the specific sensitive parameter "GW_DELAY" which control the delay time or drainage time of the overlying geologic formations; for the Huaihe River, the specific parameter was "GW_REVAP" which define the amount of water moving into the soil zone from the shallow aquifer; for Fujiang river, the specific sensitive parameter was parameter "CANMX" which control the canopy storage of water. The definition of parameters showed in Table S2.

Please also note the supplement to this comment:
https://www.hydrol-earth-syst-sci-discuss.net/hess-2018-448/hess-2018-448-AC1-supplement.pdf

**Supplement:**

**Table S1. The differences in annual mean temperature and precipitation based on WFD and meteorological observations during 1961-2001.**

| River | Annual precipitation | | | Annual mean temperature | | |
|---|---|---|---|---|---|---|
| | OBS(mm) | WFD (mm) | Difference (%) | OBS(℃) | WFD(℃) | Difference (℃) |
| Shiyang | 246.1 | 282.1 | 14.6 | 5.2 | 2.7 | -2.5 |
| Chaobai | 570.7 | 476.5 | -20.0 | 9.2 | 5.1 | -4.1 |
| Huaihe | 917.6 | 898.7 | -2.1 | 14.9 | 14.8 | -0.1 |
| Fujiang | 906.0 | 894.6 | -1.3 | 16.5 | 15.6 | -0.9 |

[Figure]

**Figure S1. The differences in monthly mean temperature and monthly precipitation based on WFD and meteorological observations during 1961-2001.**

**Table S2. The agreements in annual mean, maximum and minimum temperature, and mean annual precipitation based on WFD and downscaling climate data from five GCMs for during 1961-2001 for the four river basins.**

| River | GFDL-ESM2M | HadGEM2-ES | IPSL-CM5A-LR | MIROC-ESM-CHEM | NorESM1-M |
|---|---|---|---|---|---|
| | Difference in mean annual temperature (℃) | | | | |
| Shiyang | -0.01 | -0.03 | 0.02 | -0.00 | -0.03 |
| Chaobai | -0.01 | -0.02 | 0.08 | -0.03 | -0.01 |
| Huaihe | -0.01 | 0.01 | 0.07 | -0.03 | -0.05 |
| Fujiang | 0.31 | 0.31 | 0.36 | 0.33 | 0.29 |
| River | Difference in mean annual maximum temperature (℃) | | | | |
| Shiyang | 0.00 | 0.07 | 0.04 | 0.02 | 0.04 |
| Chaobai | 0.02 | 0.10 | -0.02 | 0.00 | 0.02 |
| Huaihe | 0.07 | 0.13 | 0.03 | 0.01 | 0.06 |
| Fujiang | 0.24 | 0.29 | 0.25 | 0.23 | 0.27 |
| | Difference in mean annual minimum temperature (℃) | | | | |
| Shiyang | -0.01 | 0.03 | 0.01 | -0.01 | 0.01 |
| Chaobai | -0.03 | 0.08 | -0.02 | -0.01 | 0.00 |
| Huaihe | 0.00 | 0.05 | -0.05 | -0.07 | -0.04 |
| Fujiang | 0.37 | 0.41 | 0.39 | 0.34 | 0.35 |
| River | Difference in mean annual precipitation (%) | | | | |
| Shiyang | 14.8 | 7.8 | 13.3 | 6.3 | 5.2 |

| | | | | | |
|---|---|---|---|---|---|
| Chaobai | 9.7 | 8.2 | 9.1 | 8.0 | 6.3 |
| Huaihe | 4.9 | 5.4 | 5.3 | 3.9 | 4.8 |
| Fujiang | 11.0 | 5.6 | 8.7 | 10.4 | 7.2 |

[Figure]

**Figure S2. The agreements in monthly mean temperature and mean precipitation based on WFD and downscaling climate data from five GCMs for during 1961-2001 for the four river basins.**

**Table S6. The agreements in mean annual runoff and evapotranspiration based on WFD and downscaling climate simulation from 5 GCMs for during 1961-2001 for the four river basins.**

| River | GFDL-ESM2M | HadGEM2-ES | IPSL-CM5A-LR | MIROC-ESM-CHEM | NorESM1-M |
|---|---|---|---|---|---|
| | Difference in mean annual runoff (%) | | | | |
| Shiyang | 16.2 | 25.3 | 16.6 | 14.4 | 12.7 |
| Chaobai | -19.3 | 21.5 | 0.5 | -9.1 | -2.3 |
| Huaihe | -7.2 | 23.7 | 9.3 | 6.3 | 3.8 |
| Fujiang | -6.2 | -16.7 | 6.3 | 0.0 | -4.3 |
| River | Difference in mean annual evapotranspiration (%) | | | | |
| Shiyang | -3.3 | -37.7 | -4.7 | -19.8 | -17.6 |
| Chaobai | 12.4 | -0.5 | 6.6 | 7.5 | 3.2 |
| Huaihe | -1.8 | -8.1 | 0.8 | 3.2 | 4.8 |
| Fujiang | 15.5 | 13.4 | 4.7 | 11.7 | 12.7 |

[Figure]

**Figure S3. The agreements in simulated mean monthly runoff and mean monthly evapotranspiration based on WFD and downscaling climate data from 5 GCMs during 1961-2001 for the four river basins.**

**Table S5.The mean of middle-year of the 30-year samples for all GCMs under RCPs and under 1.5℃ or 2℃ global warming scenarios.**

| threshold | RCP2.6 | RCP4.5 | RCP6.0 | RCP8.5 |
|---|---|---|---|---|
| 1.5℃ | 2029 | 2030 | 2032 | 2025 |
| 2.0℃ | × | 2049 | 2053 | 2038 |

**Table S3. Sensitivity results for pre-define parameters by SWAT for the four river basins**

| Rank | Shiyang River | Chaobai River | Huaihe River | Fujiang River |
|---|---|---|---|---|
| 1 | ALPHA_BF | CN2 | CN2 | CN2 |
| 2 | GWQMN | ALPHA_BF | GWQMN | ESCO |
| 3 | TIMP | GW_DELAY | RCHRG_DP | SOL_AWC |
| 4 | CN2 | ESCO | ESCO | CANMX |
| 5 | SMTMP | GWQMN | SOL_AWC | GWQMN |
| 6 | SOL_AWC | CH_N | GW_REVAP | RCHRG_DP |

**Table S4. Definition of identified sensitive parameters in SWAT hydrological model for the four river basins**

| Parameters | Definition | Processes |
|---|---|---|
| ALPHA_BF | Baseflow recession constant (days) | Groundwater |
| CANMX | Maximum canopy storage (mm $H_2O$) | Runoff |
| CH_N | Manning coefficient value | Channel |
| CN2 | SCS runoff curve number for moisture condition II | Runoff |
| ESCO | Soil evaporation compensation factor | Evaporation |
| GW_DELAY | Delay time for aquifer recharge (days) | Groundwater |
| GW_REVAP | Groundwater "Revap" coefficient (days) | Groundwater |
| GWQMN | Threshold water level in shallow aquifer for base flow (mm) | Soil |
| RCHRG_DP | Deep aquifer percolation coefficient (fraction) | Groundwater |
| SMTMP | Threshold temperature for snow melt (℃) | Snow |
| SOL_AWC | Soil available water capacity (mm/mm soil) | Soil |
| TIMP | Snow temperature lag factor | Snow |

---

## Author Comment (AC2) · 28 Dec 2018

We appreciate the Referee #3's comments and suggestions on our manuscript. We have attempted to address every point raised by them. Our responses are as follows. Anonymous Referee #3: General comment: This study attempts to investigate climate change impact on river runoff in four river basins across China, using hydrological model simulations forced by meteorological data representing 1.5 and 2 C global warming based on 5 global climate models (GCMs) under 4 emission scenarios (RCPs). The objective is further to quantify the uncertainties in the projected changes given by the GCMs and RCPs.There are a couple of general problems in the study

that need to be addressed in order to be accepted: Comment 1: There is very little information about how the hydrological model was calibrated. Which parameters were calibrated, and which criteria were used for the calibration? The inconsistent response in river runoff to the increasing precipitations over the study basins suggests that the results are strongly controlled by changes in evapotranspiration (as a result of changes in temperature and water availability). Thus we need to know more about how evapotranspiration is simulated in the model, and if and how parameters related to evapotranspiration were part of the calibration. There is also a lack of evaluation of how well the model manages to explain the observed changes in river runoff, which are referred to in the introduction. As a summary, it is doubtful if the current model is adequate for the impact study presented in the paper. Response: Thanks for this suggestion. (1) Using sensitivity analysis procedures embed in SWAT resulted in the six most sensitive parameters (Table S3) in the hydrological model for each of the four rivers and then used for model calibration. The consistent sensitive parameters among all four river basins included parameter "CN2" and "GWQMN" which control the runoff process and soil water moving process respectively. The consistent sensitive parameters for the two river basins located in the northern China was parameter "ALPHA_BF" which reflect the groundwater flow response to changes in recharge; for the two river basins located in southern China, the common sensitive parameter was "RCHRG_DP" which was a coefficient that define the aquifer percolation fraction. However, because the differences in meteorological and hydrological conditions, topography and soil properties, there was specific sensitive parameters for each river basin, such as for the Shiyang River, the specific sensitive parameters were "SMTMP" and "TIMP" which are temperature related parameters for snow; for the Chaobai River, the specific sensitive parameter "GW_DELAY" which control the delay time or drainage time of the overlying geologic formations; for the Huaihe River, the specific parameter was "GW_REVAP" which define the amount of water moving into the soil zone from the shallow aquifer; for Fujiang river, the specific sensitive parameter was parameter "CANMX" which control the canopy storage of water. The definition of parameters showed in Table

S2. (2) There is no long term ET observation available for simulated ET verify, so we compared the simulated ET based on WFD and downscaling climate data from 5 GCMs. The result showed that there are good coherence between the ET simulated based on these two kinds of dataset. The monthly distribution of ET were not changed for the most simulated runoff based on WFD and downscaling GCMs climate data. The simulated ET was underestimated for the Shiyang River, especially during the summer, with the peak of ET earlier based on the simulation of GCM HadGem2-ES. The simulated monthly ET based on GCM MIROC-ESM_CHEM also showed earlier peak in the Fujiang River. (3) The coefficient of determination (R2), Nash–Sutcliffe efficiency (Ens) were used to measures the goodness-of-fit of simulated monthly discharge with observation, and percentage of bias (Pbias) were used to evaluate systematic over- or under estimation and when the absolute value is applied it shows the magnitude the simulated monthly runoff (Green and van Griensven, 2008Moriasi et al., 2007). In general, the model simulation is considered acceptable when the Ens values are greater than 0.5, and the Pbias less than ±25% (Moriasi et al., 2007). Comment 2: A related problem is the selection of meteorological forcing data used in the study. First of all, there is no assessment presented of the agreement during the historical period between the data used for the model calibration (WFD) and the data used for the climate projections - thus we cannot assess to what extent the calibrated model us suitable for assessing the climate change impact with these data. Secondly, there is very little motivation or details given regarding the selection of the GCM models, or the selection of the 30 year periods representing 1.5 and 2.0 C warming, respectively. The selection of GCM models should be crucial for the quantification of uncertainties, which is pointed out as one of the objectives of the paper. Response: We are appreciate for the reviewer's suggestion about clarify of meteorological dataset used this research. (1) The WFD (which covers period of 1958-2001) was used to force SWAT, and also was used for bias correction of climate model outputs adopted in this study. The climate model outputs derived from Inter-Sectoral Impact Model Intercomparison Project (ISI-MIP) are spatially interpolated into 0.5° resolution and

corrected using trend-preserving bias correction approach based on WFD dataset for the period 1950–2005 for historical simulation and 2006-2099 for future projection under (Hempel et al., 2013). For subsequent hydrological projections, this study adopted downscaled climate projection data derived from the 5 GCMs and validated SWAT models and projected the impact of climate change on river runoff. The changes in averages of the annual and monthly runoff under 1.5℃ and 2.0℃global warming were compared based on the simulated runoff under all climate scenarios and with the simulated runoff based on the baseline period (1976-2005) from the five GCMs rather than the actual observed discharge data or simulated discharge forcing by WFD. This technique was used to avoid systematic errors that the SWAT model would introduce in comparing the projection period with the baseline period. Furthermore, we compared the downscaled climate data from 5 GCMs with WFD during 1961-2001. Table S6 and Figure S3 showed the agreement of WFD with the historical simulation of 5 GCMs at mean annual scale and monthly scale. The downscaled GCMs historical climate simulation showed very good agreement with WFD for both the mean annual temperature and precipitation. The differences in mean annual temperature between WFD and downscaled 5 GCMs output were -0.03℃∼0.36℃ for the four river basins, while those of mean annual maximum and minimum temperature were -0.02℃∼0.29℃ and -0.07℃∼0.41℃ respectively. There were general overestimate for mean annual precipitation based on the downscaled historical climate simulation from 5 GCMs. The difference in mean annual precipitation were 5.2%∼14.8% between WFD and downscaled historical climate simulation from 5 GCMs in the Shiyang River, those were 6.3%∼9.7% in the Chaobai River, 3.9%∼5.4% in the Huaihe River, and 5.6%∼11.0% in the Fujiang River. The downscaled GCMs historical climate simulation fitted the distribution of mean monthly temperature and precipitation with WFD very well during the 1961-2001. Generally, the downscaled GCMs output from ISI-MIP were acceptable unified set of climate drivers to allow a consistent analysis of climate change impacts on water resource at basin scale. The downscaled GCMs historical climate simulation fitted the distribution of mean monthly temperature with WFD very

well during the 1961-2001. The most of month with precipitation were overestimated by the downscaled GCMs simulation than underestimated for the four river basins, especially for the precipitation in spring and autumn. However, those differences in monthly precipitation based on WFD and downscaling climate historical simulation from five GCMs didn't change the seasonal pattern of precipitation. The downscaled GCMs output from ISI-MIP were reliable unified set of climate drivers to allow a consistent analysis of climate change impacts on water resource at basin scale. (2) The number of models contributing to CMIP5 varies with the specific experiment, but ranges from 25 to 42 for the projections under four Representative Concentration Pathway (RCP) scenarios. The large size of the CMIP5 ensemble can be particularly problematic in studies where the GCM data are used as part of a model chain including downscaling and/or impact models. However, to quantify the uncertainty associated with GCMs in climate change impact assessment, five"priority" GCMs were selected in this study recommended by ISI-MIP. The GCMs selected to span global mean temperature change and relative precipitation change as effectively as possible (Warszawski et al. 2014). The FRC index (Fractional range coverage) of the five GCMs in ISI-MIP project is 0.75 and 0.59, respectively, which is better than the five GCMs randomly selected from CMIP5, and can reasonably represent the changes of regional average temperature and precipitation (McSweeny and Jones, 2016). (3) Response: Future time horizon of global warming of 1.5°C and 2°C is derived based on 30-year running mean of global mean temperature (GMT) for each one of the 20 combinations of four RCPs (RCP2.6, RCP4.5, RC6.0 and RCP8.5) and five GCMs. When the GMT anomaly of 30-year running mean relative to pre-industrial level reaches the threshold of 1.5°C or 2°C, the 30-year window is sampled as corresponding time horizon of global warming scenario. Then year in Table S5 is estimated by averaging all center-years of the 30-year samples for all GCMs under each RCP and under each global warming scenario. Among these 20 combinations, 16 scenarios show mean GMT increases exceeding the threshold of 2°C above pre-industrial level, and 18 scenarios exceed the threshold of 1.5°C. But the changes in projected variables

(annual temperature and precipitation) are quantified relative to present day (1976 – 2005) instead of pre-industrial period in this research. We have clarify the GCMs selection and supplemented the methodology about define the 1.5℃ and 2.0℃ warming in the revised manuscript. Comment 3: In addition to the methodological and presentation issues, the paper is very uneven in the quality of the English writing, which makes it difficult to understand some of the statements. Response: Thanks for this comments. We have polished the English writing throughout the all revised manuscript and tried our best to make it more readable. Specific comments Comment 4: Figure 1: I would assume that the dark grey areas represent the study basins, but what is represented by the light grey area? I would further assume that the basin locations following the position of the surrounding graphs, but I cannot be sure without consulting the text. What is presented in the small embedded graph? It looks like some mistake. Response: Many thanks for this comment and sorry for this confusion caused by vague figure illustration. 1) The dark grey area represent the study basins, and the light grey area represent the main river basin that the study basins belonged to, and which are the Inland River Basin in northwest China (the Shiyang River), the Haihe River Basin (the Chaobai River), the Huaihe River Basin (the Huaihe River), and the Yangtze River Basin (the Fujiang River). 2) The mall embedded graph is the South China Sea Islands. These small inlands are presented in an embedded graph because it can't present at the same scale in the figure. So this is not a mistake. 3) We have marked the main river basin in Figure 1 to the location of study areas in the main river basins of China in the revised manuscript. Methodology section: Comment 5: How was the model calibrated? Which model parameters? Which objective function was used in the calibration? Response: Thanks for this suggestion. Prior to calibration, a Latin Hypercube one-at-a-time (LH-OAT) technique, proposed by Morris (1991), and implemented in SWAT-CUP (SWAT Calibration and Uncertainty Programs) was applied to investigate the sensitivity of parameters.. Using sensitivity analysis procedures embed in SWAT resulted in the six most sensitive parameters (Table S1) in the hydrological model for each of the four rivers and then these sensitive

parameters were used for model calibration. Sequential Uncertainty Fitting (SUFI2) algorithm (Abbaspour., et al., 2007) in SWAT-CUP generic interface was applied for automatic calibration and parameter optimization in the Chaobai River (Hao., et al., 2018), and manual calibration of model parameters were application for the Shiyang, Huaihe (Wang et al., 2018), and Fujiang River. The objective function used in model calibration is the Nash–Sutcliffe efficiency with the threshold of greater than 0.5. We have supplemented the method of sensitive parameters analysis and model calibration of SWAT in the four river basins in the revised manuscript. Comment 6: Please give some more explanation how the 30 year periods were selected for the different global warming thresholds - as well as how the standard deviations referring to the GCMs and the RCPs, separately were quantified. How was the standard deviation originating from the GCMs and the RCPs aggregated into the values presented in Table3? Response: Thanks for this suggestion. (1) Future time horizon of global warming of 1.5°C and 2°C is derived based on 30-year running mean of global mean temperature (GMT) for each one of the 20 combinations of four RCPs (RCP2.6, RCP4.5, RC6.0 and RCP8.5) and five GCMs. When the GMT anomaly of 30-year running mean relative to pre-industrial level reaches the threshold of 1.5°C or 2°C, the 30-year window is sampled as corresponding time horizon of global warming scenario. Then year in Table S3 is estimated by averaging all center-years of the 30-year samples for all GCMs under each RCP and under each global warming scenario. Among these 20 combinations, 16 scenarios show mean GMT increases exceeding the threshold of 2°C above pre-industrial level, and 18 scenarios exceed the threshold of 1.5°C. But the changes in projected variables (annual temperature and precipitation) are quantified relative to present day (1976 – 2005) instead of pre-industrial period in this research. (2) The uncertainty caused by RCPs was estimating using standard deviation of the mean of all GCMs under 1.5âĐČ and 2.0âĐČ global warming respectively, and the uncertainty constrained by GCMs was estimated using standard deviations of all RCPs under the two threshold of global warming, whereas the all source of uncertainty of climate change scenarios was estimating using the standard deviation of all the 18

and 16 climate scenarios under 1.5℃ and 2.0℃ global warming. (3) We have supplemented the methodology about define the 1.5℃ and 2.0℃ warming in the revised manuscript. Results section: Comment 7: I would prefer not to use sentences that only refer to a table or a figure without describing any of the results. Describe the result in the text and use the tables and figures as support. For instance, I would recommend to refer more directly to the specific results in Table 3 that supports the various statements in section 4.1. Response: Many thanks for this suggestion and this will helpful for improve my scientific wringing. I have revised the manuscript and describe the result in the text by using the information included in the table sand figures.

Please also note the supplement to this comment:
https://www.hydrol-earth-syst-sci-discuss.net/hess-2018-448/hess-2018-448-AC2-supplement.pdf
* * *
[Figure]

**Supplement:**

**Table S3. Sensitivity results for pre-define parameters by SWAT for the four river basins**

| Rank | Shiyang River | Chaobai River | Huaihe River | Fujiang River |
|------|---------------|---------------|--------------|---------------|
| 1 | ALPHA_BF | CN2 | CN2 | CN2 |
| 2 | GWQMN | ALPHA_BF | GWQMN | ESCO |
| 3 | TIMP | GW_DELAY | RCHRG_DP | SOL_AWC |
| 4 | CN2 | ESCO | ESCO | CANMX |
| 5 | SMTMP | GWQMN | SOL_AWC | GWQMN |
| 6 | SOL_AWC | CH_N | GW_REVAP | RCHRG_DP |

**Table S4. Definition of identified sensitive parameters in SWAT hydrological model for the four river basins**

| Parameters | Definition | Processes |
|------------|------------|-----------|
| ALPHA_BF | Baseflow recession constant (days) | Groundwater |
| CANMX | Maximum canopy storage (mm $H_2O$) | Runoff |
| CH_N | Manning coefficient value | Channel |
| CN2 | SCS runoff curve number for moisture condition II | Runoff |
| ESCO | Soil evaporation compensation factor | Evaporation |
| GW_DELAY | Delay time for aquifer recharge (days) | Groundwater |
| GW_REVAP | Groundwater "Revap" coefficient (days) | Groundwater |
| GWQMN | Threshold water level in shallow aquifer for base flow (mm) | Soil |
| RCHRG_DP | Deep aquifer percolation coefficient (fraction) | Groundwater |
| SMTMP | Threshold temperature for snow melt (℃) | Snow |
| SOL_AWC | Soil available water capacity (mm/mm soil) | Soil |
| TIMP | Snow temperature lag factor | Snow |

[Figure]

**Figure S3. The agreements in simulated mean monthly runoff and mean monthly evapotranspiration based on WFD and downscaling climate data from 5 GCMs during 1961-2001 for the four river basins.**

**Table S2. The agreements in annual mean, maximum and minimum temperature, and mean annual precipitation based on WFD and downscaling climate data from five GCMs for during 1961-2001 for the four river basins.**

| River | GFDL-ESM2M | HadGEM2-ES | IPSL-CM5A-LR | MIROC-ESM-CHEM | NorESM1-M |
|-------|------------|------------|--------------|----------------|-----------|

|  | Difference in mean annual temperature (℃) | | | | |
|---|---|---|---|---|---|
| Shiyang | -0.01 | -0.03 | 0.02 | -0.00 | -0.03 |
| Chaobai | -0.01 | -0.02 | 0.08 | -0.03 | -0.01 |
| Huaihe | -0.01 | 0.01 | 0.07 | -0.03 | -0.05 |
| Fujiang | 0.31 | 0.31 | 0.36 | 0.33 | 0.29 |
| River | Difference in mean annual maximum temperature (℃) | | | | |
| Shiyang | 0.00 | 0.07 | 0.04 | 0.02 | 0.04 |
| Chaobai | 0.02 | 0.10 | -0.02 | 0.00 | 0.02 |
| Huaihe | 0.07 | 0.13 | 0.03 | 0.01 | 0.06 |
| Fujiang | 0.24 | 0.29 | 0.25 | 0.23 | 0.27 |
|  | Difference in mean annual minimum temperature (℃) | | | | |
| Shiyang | -0.01 | 0.03 | 0.01 | -0.01 | 0.01 |
| Chaobai | -0.03 | 0.08 | -0.02 | -0.01 | 0.00 |
| Huaihe | 0.00 | 0.05 | -0.05 | -0.07 | -0.04 |
| Fujiang | 0.37 | 0.41 | 0.39 | 0.34 | 0.35 |
| River | Difference in mean annual precipitation (%) | | | | |
| Shiyang | 14.8 | 7.8 | 13.3 | 6.3 | 5.2 |
| Chaobai | 9.7 | 8.2 | 9.1 | 8.0 | 6.3 |
| Huaihe | 4.9 | 5.4 | 5.3 | 3.9 | 4.8 |
| Fujiang | 11.0 | 5.6 | 8.7 | 10.4 | 7.2 |

[Figure]

**Figure S2. The agreements in monthly mean temperature and mean precipitation based on WFD and downscaling climate data from five GCMs for during 1961-2001 for the four river basins.**

**Table S5. The mean of middle-year of the 30-year samples for all GCMs under RCPs and under 1.5℃ or 2℃ global warming scenarios.**

| threshold | RCP2.6 | RCP4.5 | RCP6.0 | RCP8.5 |
|---|---|---|---|---|
| 1.5℃ | 2029 | 2030 | 2032 | 2025 |
| 2.0℃ | × | 2049 | 2053 | 2038 |

---

## Author Comment (AC3) · 28 Dec 2018

Anonymous Referee #2: We appreciate the Referee #1's comments and suggestions on our manuscript. The following are our point-point replies, with reference to the order of the comments by the reviewer. Comment 1: There are some fundamental problems with this paper that make me very uneasy about recommending it for publication. Firstly, the choice of five GCMs from a much large available set of AR5 projections needs to be justified. At the very least, we need to know why these five were chosen and whether they differ from the larger set in terms of their future projections. I'd also like to know how these five perform under historical conditions. Related to this, we

need to know how the historical projections from the GCMs compare with the historical data used to calibrate the model. Section 3.2 is very unclear about this. If the historical GCM data is wildly different from the historical calibration data, I cannot see how it can be used to assess current conditions and therefore used to assess projected changes. Similarly, the SWAT calibration statistics are rather poor. The biases in calibration of 16% and 25% are much greater than the projected changes in runoff. How can we have any confidence in these projected changes when the calibrations are unable to get even the correct volume of runoff? Using the model to project seasonal changes when historical seasonal statistics were not examined is also unacceptable. The inadequacy of the model for use in climate change studies is re-iterated by the -11 to +18% change in precipitation for the Shiyang River leading to reductions in annual flow of 10% to 60%. This is not credible, and clearly the model is giving too much weighting to the impact of increases in PET. There is some value in the estimates of changes in temperature and precipitation across the four river basins, but the large bias in the hydrological model calibrations means that I cannot see how these changes in precipitation can be converted into changes in even annual runoff. Also, the changes in temperature and precipitation are predicated on just five GCMs, and we would need to know where these fall within the range of all GCMs in AR5. The authors have pretty much ignored the very large body of work emanating from Australia, the US and Europe on estimating impacts of climate change on water availability. I'd strongly suggest they go back and read the approaches that have been used elsewhere and modify their approach based on this. Response: We are appreciate for the reviewer's all comments. This study followed the top-down methodology that common used in IPCC AR4 and AR5 WGII report. Within the IPCC AR 4 and AR5 water sector, most hydrological projection studies use the precipitation and temperature downscaled from GCMs to driven hydrological models. This study adopted climate projection information derived from Inter-Sectorial Impact Model Intercomparison Project (ISIMIP). Several publication from US and Europe have by add in the revised manuscript. For the detailed information about dataset and methodology used in this study, we responded in the following parts separately,

and more information have been involved in the revised manuscript.

Specific comments: From the abstract, it appears as if the focus of the paper is on the impact of an additional 0.5 degrees global warming, not the impact of 1.5 and 2.0 degrees compared to current conditions. However, the paper does not focus on this 0.5 degree difference. Response: this research analysis the impact of 1.5 and 2.0 degrees warming compared to present day and also discuss the difference caused by the 0.5°C more warming. We have clarified this objective in the introduction part in our manuscript. Line 6. The target of 1.5 degrees is thought to be the one which might limit dangerous climate change impacts, not 2 degrees as proposed here. In fact, the comparison of 2 to 1.5 degree warming can be considered to be 'what if' we don't manage to keep to 1.5 degrees of warming? What might the additional 0.5 degrees do? That could be a useful focus of the paper, however the problems raised above mean that this cannot be done with the current approach. Response: The Paris Agreement central aim is to strengthen the global response to the threat of climate change by keeping a global temperature rise this century well below 2 degrees Celsius above pre-industrial levels and to pursue efforts to limit the temperature increase even further to 1.5 degrees Celsius (UNFCCC, 2015) So, we take 1.5 and 2.0°C as the thresholds. This study followed the methodology and Inter-Sectorial Impact Model Intercomparison Project (ISIMIP), the publication of ISI-MIP have contributed to IPCC 1.5 Special report. We developed the work main keen to give a picture how the water resource will changes under 1.5 and 2.0°C global warming , how the difference in impacts will caused by additional 0.5 degree warming, and the result will useful for decision maker of China. Comment 1: Considering the enormous range of projected changes across the AR5 GCMs, the reader needs to know why the authors selected the five GCMs used in this study. Were they just more accessible? How does the range of projections from those GCMs compare to the larger set of GCMs in AR5? Without knowing this, we have no idea if these projected changes represent a wetter/drier hotter/cooler part of the spectrum of future climate change projections. Response: We are appreciate for the reviewer's consideration about GCMs selection

in this research. The number of models contributing to CMIP5 varies with the specific experiment, but ranges from 25 to 42 for the projections under four Representative Concentration Pathway (RCP) scenarios. The large size of the CMIP5 ensemble can be particularly problematic in studies where the GCM data are used as part of a model chain including downscaling and/or impact models. However, to quantify the uncertainty associated with GCMs in climate change impact assessment, five"priority" GCMs were selected in this study recommended by ISI-MIP. The GCMs selected to span global mean temperature change and relative precipitation change as effectively as possible (Warszawski et al. 2014). The FRC index (Fractional range coverage) of the five GCMs in ISI-MIP project is 0.75 and 0.59, respectively, which is better than the five GCMs randomly selected from CMIP5, and can reasonably represent the changes of regional average temperature and precipitation (McSweeny and Jones, 2016). Comment 2: One of the key issues in hydrological modelling studies is whether the model is able to represent the current conditions well enough to be able to be used in climate change studies. In this paper, the authors claim that the model calibration and validation results are 'satisfactory'. While this may be true to some extent for the Huiaihe and Fujiang Rivers, the calibration and validation statistics for the other rivers are poor at best (remembering that they are only attempting to produce monthly, not daily streamflow). Even more concerning however in a study such as this one is that the calibration bias is 25% for the Baihe and 16% for the Huaihe River. As the projected change in annual runoff is much less than that, I cannot see how the authors can justify using such a poor calibration. I am not familiar with the WFD climate data, but I strongly suspect that is the main reason the calibrations are so poor. Are there any other datasets (local precipitation for example) that could be used instead? Also, was SWAT run on a monthly or daily basis? No information is provided. It is not at all clear which precipitation data were used to drive the SWAT model under the future climate scenarios. Section 3.2 is confusing and not at all clear. Did the authors simply take the precipitation from the climate models directly and run SWAT for both the historic and future scenarios? If so, how did these precipitation

projections, particularly the historical projections' compare to those used in the historic calibration? If they were significantly different, this gives us some information about how well the GCM's are predicting historical conditions and some confidence (or likely not) in their use in the future projections. Response: Many thanks for these suggestions which really helpful for us to revised the manuscript. (1) The WFD combined the daily statistics of ERA-40 with the monthly mean characteristics of CRU and GPCC datasets and represented a complete gridded observational dataset for bias correction of global climate data over land. WFD has been compared with CRU data and GPCC data for monthly temperature, wet days, and precipitation totals etc. WFD is considered an acceptable dataset for forcing hydrological models in comparison with gridded observation database at global scale (Essou et al., 2016). Furthermore, WFD has been used in climate change impact assessment at regional or catchment scale in China (Hao. et al., 2018; Liu et al., 2017; Chen et al, 2017; Su et al., 2017). (2) I agreed with the reviewer's comment that high resolution climate forcing or situ based observation maybe improve the hydrological model performance. The purposes of using WFD to force the SWAT hydrological model in this study: (i) to avoid the uncertainty caused by the inhomogeneity of the spatial distribution of meteorological stations, (ii) to allow a consistent analysis of climate change impacts on water resource at basin scale; (iii) to provide the case study for global comparison under ISI-MIP project. I prefer to do further investigate using available high resolution climate forcing to calibrate and validate SWAT hydrological model the four river basins to compare the hydrological model performance with forcing with 0.5 degree WFD dataset in the future research. (3) Furthermore, whether the hydrological model structure can reflect the specific hydrological process is the key factor to determine the hydrological model performance. There was a few cases showed that SWAT could be used in snowmelt-dominated streamflow (Wang and Melesse, 2005; Tolston and Shoemaker, 2007; Grusson et al., 2015), a few previous researches have indicate that SWAT model did not adequately predict winter flows or snowmelt-dominated runoff in several watershed (Peterson and Hamlett, 1998; Srivastava et al., 2006;

Chanasyk et al., 2003; Benaman et al., 2005) , which could be one reason that the low values of the Nash-Sutcliffe efficiency for the Shiyang and Chaobai rivers in the northern China with cold winter. We have explained the reason for the low values of the Nash-Sutcliffe efficiency for Shiyang and Chaibai rivers in the revised manuscript. (4) The simulations using the SWAT model were forced by WFD climate data, and they were spun-up for the period 1958–1960. The SWAT models were run at daily step and were then calibrated for the 1961–1990 and validated for 1991–2001 using monthly river runoff data from the gauging stations of the four basins. The WFD (which covers period of 1958-2001) was used to force SWAT, and also was used for bias correction of climate model outputs adopted in this study. The climate model outputs derived from Inter-Sectoral Impact Model Intercomparison Project (ISI-MIP) are spatially interpolated into $0.5°$ resolution and corrected using trend-preserving bias correction approach based on WFD dataset for the period 1950–2005 for historical simulation and 2006-2099 for future projection under (Hempel et al., 2013). For subsequent hydrological simulation, this study adopted downscaled GCMs data derived from five GCMs and validated SWAT models, and projected the impact of climate change on river runoff. he changes in averages of the annual and monthly runoff under 1.5°C and 2.0°Cglobal warming were compared based on the simulated runoff under all climate scenarios and with the simulated runoff based on the baseline period (1976-2005) from the five GCMs rather than the actual observed discharge data or simulated discharge forcing by WFD. This technique was used to avoid systematic errors that the SWAT model would introduce in comparing the projection period with the baseline period. (5) Furthermore, we compared the downscaled climate data from 5 GCMs with WFD during 1961-2001. Table S6 and Figure S3 showed the agreement of WFD with the historical simulation of 5 GCMs at mean annual scale and monthly scale. The downscaled GCMs historical climate simulation showed very good agreement with WFD for both the mean annual temperature and precipitation. The differences in mean annual temperature between WFD and downscaled 5 GCMs output were -0.03°C∼0.36°C for the four river basins, while those of mean annual maximum

and minimum temperature were -0.02°C∼0.29°C and -0.07°C∼0.41°C respectively. There were general overestimate for mean annual precipitation based on the downscaled historical climate simulation from 5 GCMs. The difference in mean annual precipitation were 5.2%∼14.8% between WFD and downscaled historical climate simulation from 5 GCMs in the Shiyang River, those were 6.3%∼9.7% in the Chaobai River, 3.9%∼5.4% in the Huaihe River, and 5.6%∼11.0% in the Fujiang River. The downscaled GCMs historical climate simulation fitted the distribution of mean monthly temperature and precipitation with WFD very well during the 1961-2001. Generally, the downscaled GCMs output from ISI-MIP were acceptable unified set of climate drivers to allow a consistent analysis of climate change impacts on water resource at basin scale. The downscaled GCMs historical climate simulation fitted the distribution of mean monthly temperature with WFD very well during the 1961-2001. The most of month with precipitation were overestimated by the downscaled GCMs simulation than underestimated for the four river basins, especially for the precipitation in spring and autumn. However, those differences in monthly precipitation based on WFD and downscaling climate historical simulation from five GCMs didn't change the seasonal pattern of precipitation. The downscaled GCMs output from ISI-MIP were reliable unified set of climate drivers to allow a consistent analysis of climate change impacts on water resource at basin scale. Commnent 2: Figure 2 shows that all future projections for the Shiyang River are for reductions in annual flow (of between 10% and 60%), but Table 3 states that annual precipitation shows a range of changes from an 11% decrease to an 18% increase. If the modeling indicates that an 18% increase in precipitation will lead to a reduction in annual runoff then the model is clearly inadequate for use in climate change studies. Response: Many thanks for this comment. We have noticed this issue in this research, and have discussed the possible reason of reduction in annual runoff. Precipitation is the main input of surface water resources and evapotranspiration (ET) is the main output. Ma et al. (2008) indicated that decreased precipitation and increased potential ET contribute most to the reduction of streamflow in northwest China. In Shiyang River under 1.5 and

2.0°C global warming, the projected ensemble mean annual precipitation increased 3% and 5%, and range of changes from 11% decrease to 18% increase. While, the simulated change of ET in the Shiyang River showed robust increase of 22% and 14%, and range of changes from 3% decrease to 52% increase. This implies the increase in simulated ET contributes most to the decrease in simulated annual runoff in the Shiyang River. As mentioned in SWAT model calibration and validation part, a few previous researches have indicate that SWAT model did not adequately predict winter flows or snowmelt-dominated runoff in several watershed, which could be one reason that the low values of the Nash-Sutcliffe efficiency for the Shiyang. Moreover, Li et al. (2016) indicated that frozen soil meltwater accounted for about 20% of river runoff during the flood season, while glacier meltwater contributed only about 3% in the Shiyang River. However, the glacier meltwater process was not considered in SWAT-based simulations in this study, which would have also contributed to the decrease in simulated annual runoff in the Shiyang River. These all induce high uncertainties using SWAT for hydrological simulation in the Shiyang River. We have clarify this finding in the revised manuscript. Comment: While it is written well overall, considering the authors all presumably have English as a second language, there are a few sentences that do not make sense, such as line 23-25. Response: we have revised this sentence to "For the region with simulated water resource declined, the uncertainties in simulated runoff usually constrained by global hydrological models." Figure 1. What does the light grey shading signify? What is the inset attempting to show? Response: Many thanks for this comment and sorry for this confusion caused by vague figure illustration. The dark grey area represent the study basins, and the light grey area represent the main river basin that the study basins belonged to, and which are the Inland River Basin in northwest China (the Shiyang River), the Haihe River Basin (the Chaobai River), the Huaihe River Basin (the Huaihe River), and the Yangtze River Basin (the Fujiang River). We have marked the main river basin in Figure 1 to the location of study areas in the main river basins of China in the revised manuscript.

Please also note the supplement to this comment:
https://www.hydrol-earth-syst-sci-discuss.net/hess-2018-448/hess-2018-448-AC3-supplement.pdf

[Figure]

**Supplement:**

[revised manuscript text omitted]
. However, the complex terrain in different river basins makes it difficult for reanalysis data to reach satisfactory agreement with station based observation. Our study show both underestimation and overestimation in precipitation and temperature. This could induce the uncertainty in the river runoff simulation. However, previous research indicates that the gridded climate dataset can be used in hydrological modeling, and the performance of hydrological model will improve by model calibration and validation (Xu et al., 2011). Furthermore, the SWAT hydrological model calibrated and validated based on WFD, then drive 
[revised manuscript text omitted]
 | **1.8** | 2.2 | 1.3 | 0.28 | 0.17 | 0.10 | **0** | 10 | -6 | 4.6 | 4.1 | 2.1 |

[Figure]

**Figure S1. The differences in monthly mean temperature and monthly precipitation based on WFD and meteorological observations during 1961-2001.**

[Figure]

**Figure S2. The agreements in monthly mean temperature and mean precipitation based on WFD and downscaling climate data from five GCMs for during 1961-2001 for the four river basins.**

[Figure]

**Figure S3. The agreements in simulated mean monthly runoff and mean monthly evapotranspiration based on WFD and downscaling climate data from 5 GCMs during 1961-2001 for the four river basins.**

**Table S1. The differences in annual mean temperature and precipitation based on WFD and meteorological observations during 1961-2001.**

| River | Annual precipitation | | | Annual mean temperature | | |
|---|---|---|---|---|---|---|
| | OBS(mm) | WFD (mm) | Difference (%) | OBS(℃) | WFD(℃) | Difference (℃) |
| Shiyang | 246.1 | 282.1 | 14.6 | 5.2 | 2.7 | -2.5 |
| Chaobai | 570.7 | 476.5 | -20.0 | 9.2 | 5.1 | -4.1 |
| Huaihe | 917.6 | 898.7 | -2.1 | 14.9 | 14.8 | -0.1 |
| Fujiang | 906.0 | 894.6 | -1.3 | 16.5 | 15.6 | -0.9 |

**Table S2. The agreements in annual mean, maximum and minimum temperature, and mean annual precipitation based on WFD and downscaling climate data from five GCMs for during 1961-2001 for the four river basins.**

| River | GFDL-ESM2M | HadGEM2-ES | IPSL-CM5A-LR | MIROC-ESM-CHEM | NorESM1-M |
|---|---|---|---|---|---|
| | Difference in mean annual temperature (℃) | | | | |
| Shiyang | -0.01 | -0.03 | 0.02 | -0.00 | -0.03 |
| Chaobai | -0.01 | -0.02 | 0.08 | -0.03 | -0.01 |
| Huaihe | -0.01 | 0.01 | 0.07 | -0.03 | -0.05 |
| Fujiang | 0.31 | 0.31 | 0.36 | 0.33 | 0.29 |
| River | Difference in mean annual maximum temperature (℃) | | | | |
| Shiyang | 0.00 | 0.07 | 0.04 | 0.02 | 0.04 |
| Chaobai | 0.02 | 0.10 | -0.02 | 0.00 | 0.02 |
| Huaihe | 0.07 | 0.13 | 0.03 | 0.01 | 0.06 |
| Fujiang | 0.24 | 0.29 | 0.25 | 0.23 | 0.27 |
| | Difference in mean annual minimum temperature (℃) | | | | |
| Shiyang | -0.01 | 0.03 | 0.01 | -0.01 | 0.01 |
| Chaobai | -0.03 | 0.08 | -0.02 | -0.01 | 0.00 |
| Huaihe | 0.00 | 0.05 | -0.05 | -0.07 | -0.04 |
| Fujiang | 0.37 | 0.41 | 0.39 | 0.34 | 0.35 |
| River | Difference in mean annual precipitation (%) | | | | |
| Shiyang | 14.8 | 7.8 | 13.3 | 6.3 | 5.2 |
| Chaobai | 9.7 | 8.2 | 9.1 | 8.0 | 6.3 |
| Huaihe | 4.9 | 5.4 | 5.3 | 3.9 | 4.8 |
| Fujiang | 11.0 | 5.6 | 8.7 | 10.4 | 7.2 |

**Table S3. Sensitivity results for pre-define parameters by SWAT for the four river basins**

| Rank | Shiyang River | Chaobai River | Huaihe River | Fujiang River |
|------|---------------|---------------|--------------|---------------|
| 1 | ALPHA_BF | CN2 | CN2 | CN2 |
| 2 | GWQMN | ALPHA_BF | GWQMN | ESCO |
| 3 | TIMP | GW_DELAY | RCHRG_DP | SOL_AWC |
| 4 | CN2 | ESCO | ESCO | CANMX |
| 5 | SMTMP | GWQMN | SOL_AWC | GWQMN |
| 6 | SOL_AWC | CH_N | GW_REVAP | RCHRG_DP |

**Table S4. Definition of identified sensitive parameters in SWAT hydrological model for the four river basins**

| Parameters | Definition | Processes |
|------------|------------|-----------|
| ALPHA_BF | Baseflow recession constant (days) | Groundwater |
| CANMX | Maximum canopy storage (mm $H_2O$) | Runoff |
| CH_N | Manning coefficient value | Channel |
| CN2 | SCS runoff curve number for moisture condition II | Runoff |
| ESCO | Soil evaporation compensation factor | Evaporation |
| GW_DELAY | Delay time for aquifer recharge (days) | Groundwater |
| GW_REVAP | Groundwater "Revap" coefficient (days) | Groundwater |
| GWQMN | Threshold water level in shallow aquifer for base flow (mm) | Soil |
| RCHRG_DP | Deep aquifer percolation coefficient (fraction) | Groundwater |
| SMTMP | Threshold temperature for snow melt (°C) | Snow |
| SOL_AWC | Soil available water capacity (mm/mm soil) | Soil |
| TIMP | Snow temperature lag factor | Snow |

**Table S5.The mean of middle-year of the 30-year samples for all GCMs under RCPs and under 1.5℃ or 2℃ global warming scenarios.**

| threshold | RCP2.6 | RCP4.5 | RCP6.0 | RCP8.5 |
|-----------|--------|--------|--------|--------|
| 1.5℃ | 2029 | 2030 | 2032 | 2025 |
| 2.0℃ | × | 2049 | 2053 | 2038 |

**Table S6. The agreements in mean annual runoff and evapotranspiration based on WFD and downscaling climate simulation from 5 GCMs for during 1961-2001 for the four river basins.**

| River | GFDL-ESM2M | HadGEM2-ES | IPSL-CM5A-LR | MIROC-ESM-CHEM | NorESM1-M |
|-------|------------|------------|--------------|----------------|-----------|

|  | Difference in mean annual runoff (%) | | | | |
|---|---|---|---|---|---|
| Shiyang | 16.2 | 25.3 | 16.6 | 14.4 | 12.7 |
| Chaobai | -19.3 | 21.5 | 0.5 | -9.1 | -2.3 |
| Huaihe | -7.2 | 23.7 | 9.3 | 6.3 | 3.8 |
| Fujiang | -6.2 | -16.7 | 6.3 | 0.0 | -4.3 |
| River | Difference in mean annual evapotranspiration (%) | | | | |
| Shiyang | -3.3 | -37.7 | -4.7 | -19.8 | -17.6 |
| Chaobai | 12.4 | -0.5 | 6.6 | 7.5 | 3.2 |
| Huaihe | -1.8 | -8.1 | 0.8 | 3.2 | 4.8 |
| Fujiang | 15.5 | 13.4 | 4.7 | 11.7 | 12.7 |

---

## Author Response (AR1)

Comments to the Author:

I have received three reviews of the manuscript.

The first Referee provided no substantial comments on the paper. Both the second and the third Referees provided extensive comments and recommended re-considering the manuscript after major revisions.

A few comments were properly addressed by the authors in their responses. However, major criticism in the Referees' reports was answered by the authors in a rather formal manner. I fully agree with the major criticism.

First of all, I would recommend the authors to pay attention to the Referees' concern regarding the used SWAT model applicability to the impact study and to provide more convincing evidence of the model performance on the basis of the historical data. In fact, the calibration statistics are poor, inconsistent response in simulated river runoff to the increasing precipitations looks questionable, etc.

Second, major problem is the selection of meteorological forcing data used in the study. I share the all Referees concern about the lack of comparison between the observed and GCMs-based data, regarding the selection of the GCM models, etc.

Overall, I recommend the authors to consider the principle criticisms and re-submit the revised manuscript for the Editor's review and final decision on opportunity of publication in HESS.

Response to editor: We appreciate the editor's effort put into this manuscript and give our opportunity to revise this manuscript. The comments and suggestions provided by editor really helpful for us to improve the current manuscript. We have attempted to address every point raised by the editor which also the major criticism in the Referees' reports in our revised manuscript and hope we have the opportunity of publication in HESS.

The followings are summary for response to the key concerns, and details showed in the revsided manuscript:

**1. The application of SWAT.**

In this study, SWAT hydrological model were calibrated based on SWAT-CUP (SWAT Calibration and Uncertainty Programs) to improve the fit between simulated and observed discharge. However, the auto-calibration didn't result in satisfactory performance of the hydrological model in the SYR and the CBR. It is thought that this is because there are some model-observation divergences within the 1961–1990 calibration period that are simply too large to be resolved by an auto-calibration routine. Following the unsuccessful application of auto-calibration routines, a more extensive manual calibration was undertaken by manually varying the six most sensitive parameters in the SWAT. Following manual calibration of model parameters, a relative satisfactory fit between observed and simulated monthly river flow was obtained in the SYR and the CBR.

In general, the model simulation is considered acceptable when the  $E_{ns}$  values are greater than 0.5,  $R^2$  should exceed 0.6, and the  $P_{bias}$  less than  $\pm 20\%$ . Model performance statistics over the calibration and validation periods were all found "satisfactory" for the four basins. The performance statistics  $E_{ns}$  and  $R^2$  were both > 0.8 and considered highly acceptable for the two basins in southern China (i.e., the HHR and the FJR) for both the calibration and the validation periods. The same performance statistics were considered reasonably acceptable for the two basins in northern China (i.e., the SYR and the CBR) with efficiencies in the range 0.58~0.82. The percentage bias was generally less than 20% (excepted for the Baihe River for the calibration period) in the four rivers.

Furthermore, we evaluated the performance of discharge simulation of SWAT by comparing the graphical plots including monthly time series which reflects the month to-month sequencing (added as Figure S3), and flow duration curve which shows the frequency distributions of discharge (Fig.S4) in the revised manuscript. The two kind of graphical plots comparison well matching between the observed and simulated discharge during 1961-2001, details in Section 3.1 and Figure S3 and Figure S4 in the revised manuscript.

**2. Inconsistent response in simulated river runoff to increasing precipitation.**

We fully understand the argument about inconsistent response in simulated river runoff to increasing precipitation in the four river basins.

We added a section 5.1 Climate change impact on runoff to compare our result with previous studies (Chen et al. 2014; Liu et al., 2012; Ma et al., 2008). The four river basins in this study represent climate from dry to wet, and the response of runoff to precipitation change also coincided with the previous findings that more increase in precipitation need to maintain runoff in drier basins.

Considering evapotranspiration (ET) is the main output of surface water resources, we added the simulated change in ET (added Figure 4). The results showed that a general increase in simulated ET in all four basins accompanied with global warming, however, the magnitude of the simulated change of ET varies across the basins, i.e., it is larger in the two basins in north China than in the two basins in south China. For the two rivers located in northern China, the simulated change of ET in the SYR shows increase of 21% and 13%, while that of the CBR shows increase of 4% and 6% under 1.5  $\$  and 2.0  $\$  warming, respectively, which implies the increase in simulated ET contributes most to the decrease in simulated annual runoff in the SYR.

1National Climate Center, China Meteorological Administration, Beijing, 100081, China
 2Chongqing Climate Center, Chongqing, 401147, China
 3Anhui Climate Center, Hefei, 230031, China
 4Anhui Meteorological Observatory, Hefei, 230031, China
 5Beijing Meteorological Disaster Prevention Center, Beijing, 100089, China

[revised manuscript text omitted]
 the Nash Sutcliffe efficiency for the Shiyang and Chaobai rivers in the northern China with cold winter. It can be summarized that appears to capture successfully the underlying hydrology of the four river basins evaluated by the three statistic metrics and compared by the monthly discharge series, and flow duration curve. The successful application of the SWAT in different climate regions is considered adequate verification of the suitability of the model for future climate change impact
- 20 on runoff in the four selected basins.

25

**3.2 Climate change projection and runoff simulation**

The future scenarios for limiting global warming thresholds of 1.5  $\$  and 2.0  $\$  were derived based on 30-year running mean of global mean temperature (GMT) followed the methodology of Liu et al. (2017) for each one of the 20 combinations under four RCPs and five GCMs of the climate projection subset. Table S5 showed the averaged middle year of the 30-year samples for all GCMs under each RCPs of 1.5  $\$  and 2.0  $\$  global warming. There were 18 scenarios under the threshold of -1.5  $\$  above preindustrial levels and 16 scenarios under the threshold of -2.0  $\$ . These scenarios were used to quantify the difference in the changes of the projected annual temperature and precipitation in the four river basins by comparing with the baseline period (1976-2005).

To indicate the overall magnitude and difference of the climate change projection under limiting global warming thresholds

30 of 1.5 ℃ and 2.0 ℃, the projected changes in mean annual temperature and annual precipitation were quantified by the value of ensemble mean under all climate scenarios (Ave.), and the projected changes in maximum and minimum annual temperature and annual precipitation (Max. and Min.) among all climate scenarios. The uncertainty caused by RCPs was estimating using standard deviation of the mean of all GCMs under 1.5℃ and 2.0℃ global warming respectively, and the

uncertainty constrained by GCMs was estimated using standard deviations of all RCPs under the two threshold of global warming, whereas the all source of uncertainty of climate change scenarios was estimating using the standard deviation of all the 18 and 16 climate scenarios under  $1.5^{\circ}$ C and  $2.0^{\circ}$ C global warming.

The hydrological simulation adopted the climate projection subset for the downscaling climate data, and the future climate

- 5 scenarios from five GCM and validated SWAT models s in the four basins, and projected the impact of climate change on river discharges. Generally, the hydrological simulations based on downscaling climate data from five GCMs for baseline period compared well with those based on WFD, and were acceptable subsequent hydrological projection (Table-Tab. S6 and Figure-Fig. S3). The changes in averages of the annual and monthly runoff were compared based on the simulated runoff under all climate scenarios and with the simulated runoff based on the baseline period (1976-2005) from the five GCMs for baseline rather than the actual observed discharge data or simulated discharge forcing by WFD.
- The simulated changes in mean annual runoff were quantified by the value of ensemble mean annual runoff of all climate scenarios under 1.5°C and 2.0°C global warming, and mean annual runoff under RCP 2.6, RCP4.5, RCP6.0 and RCP8.5 respectively, and mean annual runoff under GCM GFDL-ESM2M, HaDGem2, IPSL\_CM5A-LR, MIROC-ESM-CHEM, and NorESM1-M respectively. The simulated changes in monthly runoff were analysis by the proportion of monthly runoff in
- annual runoff using the mean of baseline period for 5 GCMs, and ensemble mean, maximum and minimum of simulated monthly runoff under all combined climate scenarios of GCMS and RCPs for 1.5 °C and 2.0 °C global warming, respectively.

**4 Results**

**4.1 Projected climate change**

The statistics of the projected climate change for the four basins from the 16-18 scenarios under 1.5 °C warming and the 18 16 scenarios under 2.0 °C warming are-were shown in Table Table. 3. The results show substantial warming for all four basins under two thresholds global warming. The projected changes in ensemble mean annual temperature show 1.5 °C increase under 1.5 °C global warming and 2.2 °C increase under 2.0 °C warming for the SYRhiyang and the Chaobai riversCBR. While, the projected changes in ensemble mean annual precipitation show 3% and 5% increase under 1.5 °C warming, and 5% and 8% increase under 2.0 °C warming for the Shiyang-SYR and the Chaobai riversCBR, respectively.
The projected changes in ensemble mean annual temperature show 1.1 °C and 1.2 °C increase under 1.5 °C warming, and 1.8 °C increase under 2.0 °C warming for the Huaihe-HHR and the Fujiang riversFJR. The projected changes in ensemble mean annual precipitation are minor for the Huaihe-HHR and Fujiang riversFJR (i.e., <±3%). All statistics for the two basins in northern China indicate generally warmer and wetter conditions in future compared with the 'present day.' The two basins in southern China are projected to have less warming and no consistent change in the projected ensemble mean annual precipitation.</li>

The greatest range in projected changes in annual mean temperature occurs in the HHRHuaihe River, with the warming range of 0.3--1.6 °C under 1.5 °C warming and that of 0.7--2.3 °C under 2.0 °C warming among all projection scenarios. The projected range in annual temperature is also large for the Shivang SYRiver, with change in the range of warming 0.9– ~2.4 °C under 1.5 °C warming and that of 1.7-2.9 °C under 2.0 °C warming, respectively. There is no consistency in the

5 direction of range in projected annual precipitation change among the four river basins, with increases ranged 10% to 20% and decreases ranged -6% to -11%. For the two river basins in southern China, the range in projected change in annual precipitation is less than for the two basins in northern China.

The uncertainty is substantial in annual precipitation projection compared with that associated with annual temperature projection, with considerable dispersion among the scenarios. Comparing the uncertainty under limiting global warming under thresholds of 1.5  $^{\circ}$ C -and 2.0  $^{\circ}$ C, the former has larger uncertainties for the projected change in annual precipitation

- than that under the later; however, it is the opposite for the projected change in annual temperature. There is generally larger uncertainty constrained by the GCMs (i.e., about 1-3 times) than associated with the RCPs for the projected annual precipitation for all four river basins. However, the uncertainty in annual temperature projection associated
- with the RCPs is larger in the Shivang-SYR River (about 2 times) and in the Huaihe RiverHR (about 1.5-23.0 times) than 15 constrained with the GCMs. All these findings show the uncertainty in the projection of annual precipitation mainly constrained by GCM structure across the four river basins, whereas the dominance of the uncertainty associated with either the GCMs or the RCPs in the projection of annual mean temperature is dependent on the basin.

**4.2 Simulated annual river runoff**

Figure 2 shows the simulated ensemble mean annual river runoff based on all combined climate scenarios, and the average 20 simulated annual river runoff of the four RCPs and the average of the five GCMs. The simulated ensemble mean annual runoff decreases for the Shiyang RiverYR by about 25% and 33% under 1.5 °C and 2.0 °C warming, respectively, and the simulated change for the Fujiang RiverJR shows a decrease of about 4% under 1.5 °C warming. The simulated ensemble mean annual river runoff shows an increase with magnitude of about 8% and 12% for the Chaobai RiverBR and about 8% and 7% for the Huaihe RiverHR under 1.5 °C and 2.0 °C warming, respectively.

- 25 The decrease in the simulated annual river runoff for the Shiyang RiverYR occurs across all the combined scenarios, ranging ranged from 0% to -72% under 1.5 °C warming and from -11% to -63% under 2.0 °C 
[revised manuscript text omitted]

Xia, B., et al.: Evolution of the significance of abrunt changes in presinitation and pupelf process in China. J. Hydrol. 560.

Xie, P., et al.: Evaluation of the significance of abrupt changes in precipitation and runoff process in China, J. Hydrol., 560, 451-460, https://doi.org/10.1016/j.jhydrol.2018.02.036, 2018.

 Xu, H., Taylor, R., Kingston, D., Jiang, T., Thompson, J., and Todd, M.: Hydrological modeling of River Xiangxi using SWAT2005: a comparison of model parameterizations using station and gridded meteorological observations, Quatern. Int., 226, 54–59, doi:10.1016/j.quaint.2009.11.037, 2010b.

Xu, K., Milliman, J.D., Xu, H.: Temporal trend of precipitation and runoff in major Chinese Rivers since 1951, Glob. Planet.
 Change., 73 (3-4), 219–232, https://doi.org/10.1016/j.gloplacha.2010.07.002, 2010a.

Xu, X., Yang, D., Yang, H., et al.: Attribution analysis based on the Budyko hypothesis for detecting the dominant cause of runoff decline in Haihe basin, J. Hydrol., 510,530-540, https://doi.org/10.1016/j.jhydrol.2013.12.052, 2014.

Yang, Y. and Tian, F.: Abrupt change of runoff and its major driving factors in Haihe River Catchment, China, J. Hydrol.,

15 374 (3-4), 373-383, https://doi.org/10.1016/j.jhydrol.2009.06.040, 2009.

Zhang, W., Pan, S., Cao, L., et al.: Changes in extreme climate events in eastern China during 1960-2013: A case study of the Huaihe River Basin, Quat. Int., 380-381, 22-34, https://doi.org/10.1016/j.quaint.2014.12.038, 2015.

Zhang, W. and Villarini, G.: Heavy precipitation is highly sensitive to the magnitude of future warming, Clim. Change, 45(1-2), 249-257, https://doi.org/10.1007/s10584-017-2079-9, 2017.

20 Zhu, Q. and Li, Y.: Environmental Restoration in the Shiyang River Basin, China: Conservation, reallocation and more efficient use of water, Quati. Procedia., 24-34, https://doi.org/10.1016/j.aqpro.2014.07.005, 2014.

Xu, H. and Luo, Y.: Climate change and its impacts on river discharge in two climate regions in China, Hydrol. Earth Syst. Sci., 19, 4609-4618, https://doi.org/10.5194/hess-19-4609-2015, 2015.

15 Figure 1. Locations and average monthly precipitation/runoff of the four selected basins in China.

---

## Author Response (AR2)

The manuscript has significantly improved over the previous version. The new figures with hydrographs and flow duration curve are informative and adds to the paper. However, there seem to be a noticeable downgrade in writing quality in the added paragraphs. I have two additional main comments and a couple of minor comments, including some examples of spelling/wording errors in the text.

Response: We appreciate all the effort that the reviewer put into this manuscript. Both the two main comments and the minor comments for spelling/words errors are valuable for us to improve the manuscript more scientific and readable. We have revised the manuscript according all the comments and polish the added paragraphs in the revised manuscript.

To make the final revision clear, we have accepted all the changes in the previous version and marked the changes according the comments in the revised manuscript. The followings are point to point response to the reviewer.

1.     The authors should be more careful in using the word "cause" without proving the causation. For example, in P11L19 they use "precipitation change would cause a change in runoff". Runoff change is the result of water balance change between key physical processes governing not only the change of precipitation but also the change of ET, groundwater, snowmelt etc. A proper discussion should investigate the change in water balances for the four basins studied and show that precipitation change is the dominating factor (if true). Note also some key processes are also missing from the model, for example the effect of elevated $CO_2$ concentration on vegetation growth stimulation and water use efficiency, therefore more caution should be applied before using the word  "cause". Later in P14L13 the authors also wrote 'cause a "wet-get-wetter" and "dry-get-drier" response'. In general, a more thorough explanation/analysis is needed to show, not tell about the causation between changes of hydrological variables.

Response: We agreed with the reviewer's comment. Precipitation is the main input of surface water resources and ET is the main output. Furthermore, the runoff change is the result of water balance including precipitation, ET, groundwater, snowmelt etc. We have analyzed the changes in precipitation, runoff in the Section 4, and also discuss the changes in ET in Section 5. Precipitation is not the only contributor of runoff changes, the water balance component dominated the changes in runoff will depend on specific hydrological processes in each basin. As we discussed in Section 5, in SYR, the increase in simulated ET is the main reason for decrease in simulated runoff, although, there is increase in projected precipitation in SYR in the arid climate region. That means more precipitation increase will need to main the unchanged in runoff under 1.5 and 2.0℃ global warming. So, we have change the word "cause" to "maintain" in L29P10 and L31P10, which is also consistent with the analysis of previous study we have cited that more increase in precipitation need to maintain runoff in drier basins.

We have changed "cause" to "coincide with" in P12L25. In this study, we summarized the finding that 0.5℃ more global warming have a "wet-get-wetter" and "dry-get-drier" response in the two basins in northern China. However, the causation between climate and hydrological variables are quite complex, we couldn't attributed the mentioned phenomena of runoff to the 0.5℃ more global warming.
we will investigate the mechanism and attribution of hydrological variables change to climate change in future research by exploring new methodology.

2. Another comment concerns the metadata information of observation data used for comparison in this study. The authors added necessary comparison between WFD and meteorological observations in Table S1 and Fig. S1, however the new observations were not described. Metadata information such as data source, time period, temporal resolution and uncertainties is necessary to improve the transparency and credibility of this study to the readers. Similarly, it would help if the values of calibrated parameters are provided in addition to names of "identified sensitive parameters" in Table S4.

Response: We have added the meteorological observations (OBS) information in P4L25-28 in the revised manuscript, and showed the comparison of WFD dataset with OBS dataset for time period 1961-2001 for mean annual and monthly temperature and precipitation. The added information will help to improve the transparency and credibility of this study to the readers.

Table S4 showed the identified sensitive parameters in the four river basins, which was summarized by the identified the six most sensitive parameters for each river basin. Considering the most of identified sensitive parameters were spatial based, such as the fitted value for SOL_AWC were unique for various soil type in each river basin, the fitted value for SMTMP in SYR were specific for different elevation, etc. So we didn't show the values of calibrated parameters in the Table S4. Instead, we indicated which parameters were consistent in all four river basins, which parameters were consistent in both two river basins located in the northern China and southern China respectively, and which were specific parameters for each basin. We think these information might useful and credible for the readers when they calibrated SWAT model in specific basin.

Minor comments:

1. In authors' reply: 2P1. "In general, the model simulation is considered acceptable when the Ens values are greater than 0.5, R2 should exceed 0.6, and the Pbias less than 20%." Is there a citation supporting this statement?

Response: According the reference "Model evaluation guidelines for systematic quantification of accuracy in watershed simulations" (Moriasi and Jones, 2007), Nash-Sutcliffe efficiency (Ens), percent bias (Pbias), and ratio of the root mean square error to the standard deviation of measured data (RSR) were recommended quantitative statistics in model evaluation. In general, model simulation can be judged as satisfactory if Ens >0.5, RSR≤0.7, and if Pbias±25% for streaflow, however, should be adjusted if measurement uncertainty is either very low or very high. The citation is showed in P6L20 in the text and listed in P15L14-15 in the reference.

2. Figure 1. which years were used to generate the seasonal cycle? Suggest adding dots to indicate the location of gauges.

Response: The time period used to generate the seasonal cycle was 1961-2000, and we have added the observation period in P3L19 in the revised manuscript. We have added the location of hydrological gauges of the four basin in Figure 1. in the revised manuscript.

3. Figure 2. The meaning of the dots (ensemble mean or median) should be explained in the

figure caption.

Response: We have explained the meaning of dots in the caption of Figure 2 in the revised manuscript.

4. Figure 4. add a-d labels for the figure panels?

Response: We have added a-d labels for the figure panels in Figure 4 in the revised manuscript.

5. Table 3. What do you mean by "Uncertainty"

Response: Forecasts of climate change are inevitably uncertain. The uncertainty comes from three sources, namely model uncertainty, scenario uncertainty and the random, internal variability of climate. Considering the challenge to address uncertainties for all sources, which we only focus on the uncertainties constrained by GCMs and RCPs in this study.

Table 3 showed the uncertainty in projected mean annual temperature and annual precipitation among all climate scenarios, among four RCPs, and among five GCMs. The uncertainty caused by RCPs was estimating using standard deviation of the mean of all GCMs (mean of five GCMs under RCP2.6, RCP4.5, RCP6.5 and RCP8.5 respectively) under 1.5℃ and 2.0℃ global warming, and the uncertainty constrained by GCMs was estimated using standard deviations of all RCPs (mean of four RCPs under GCM GFDL-ESM2M, HaDGem2, IPSL_CM5A-LR, MIROC-ESM-CHEM, and NorESM1-M respectively) under the two global warming, whereas the all source of uncertainty of climate change scenarios was estimating using the standard deviation of all the 18 and 16 climate scenarios under 1.5℃ and 2.0℃ global warming.

6. P2L27 "simulated water resource declined" -> "simulated water resource decrease"

Response: We have changes this in P2L24 in the revised manuscript.

7. P4L10 what do you mean by "reduced and intense droughts"?

Response: There a extra "and" in this phrase. We have revised this sentence to "Climate change has led to increased severe storms, decreased intense droughts in HHR Basin (Zhang et al., 2015)" in the revised manuscript.

8. P4L32 less -> "less than"

Response: We have changes this in P4L30 in the revised manuscript.

9. P5L18 "insistent well" -> "good". Note the good agreement is by design: the GCMs are bias corrected against WFD data. The authors probably have done a bit too much showing their good agreement in climatology, which should come as a surprise to very few readers.

Response: We have changes this in P5L15 in the revised manuscript.

10. P11L16 "also coincided" -> "is also consistent"

Response: We have changes this in P10L28 in the revised manuscript.

11. P7L8 "underestimate" -> "underestimation"

Response: We have changed "underestimate" to "underestimation" in P6L32-34 and P7L1 in the

revised manuscript.

12. P8L5 Is there an extra "s"?
Response: We have deleted the extra "s" in P7L23 in the revised manuscript.

13. P14L29 Is last sentence just repeating the first one in this paragraph? Suggest deletion.
Response: We have deleted the last sentence in P13L8-9 in the revised manuscript.

[revised manuscript text omitted]
 | **1.8** | 2.2 | 1.3 | 0.28 | 0.17 | 0.10 | **0** | 10 | -6 | 4.6 | 4.1 | 2.1 |

**Supplement:**

[Figure]

Figure S1. The differences in monthly mean temperature and monthly precipitation based on WFD and OBS in (a) SYR, (b) CBR, (c) HHR, and (d) FJR during 1961-2001.

[Figure]

Figure S2. The agreements in monthly mean temperature and mean precipitation based on WFD and downscaling climate data from five GCMs in (a) SYR, (b) CBR, (c) HHR, and (d) FJR during 1961-2001.

[Figure]

[Figure]

Figure S3. Observed and simulated monthly discharge during calibration period (1961-1990) and validation period (1991-2001) for (a) SYR, (b) Chaohe River, (c) Baihe River, (d) HHR, and (e) FJR.

[Figure]

Figure S4. Observed and simulated flow duration curve based on monthly discharge during 1961-2001 for (a) SYR, (b) Chaohe River, (c) Baihe River, (d) HHR, and (e) FJR.

[Figure]

Figure S5. The agreements in simulated mean monthly runoff and mean monthly evapotranspiration based on WFD and downscaling climate data from 5 GCMs during 1961-2001 for (a) SYR, (b) CBR, (c) HHR, and (d) FJR.

Table S1. The differences in annual mean temperature and precipitation based on WFD and OBS during 1961-2001 for the four river basins.

| River | Annual precipitation | | | Annual mean temperature | | |
|---|---|---|---|---|---|---|
| | OBS(mm) | WFD (mm) | Difference (%) | OBS(℃) | WFD(℃) | Difference (℃) |
| SYR | 246.1 | 282.1 | 14.6 | 5.2 | 2.7 | -2.5 |
| CBR | 570.7 | 476.5 | -20.0 | 9.2 | 5.1 | -4.1 |
| HHR | 917.6 | 898.7 | -2.1 | 14.9 | 14.8 | -0.1 |
| FJR | 906.0 | 894.6 | -1.3 | 16.5 | 15.6 | -0.9 |

Table S2. The agreements in annual mean, maximum and minimum temperature, and mean annual precipitation based on WFD and downscaling climate data from five GCMs  during 1961-2001 for the four river basins.

| River | GFDL-ESM2M | HadGEM2-ES | IPSL-CM5A-LR | MIROC-ESM-CHEM | NorESM1-M |
|---|---|---|---|---|---|
| | Difference in mean annual temperature (℃) | | | | |
| SYR | -0.01 | -0.03 | 0.02 | -0.00 | -0.03 |
| CBR | -0.01 | -0.02 | 0.08 | -0.03 | -0.01 |
| HHR | -0.01 | 0.01 | 0.07 | -0.03 | -0.05 |
| FJR | 0.31 | 0.31 | 0.36 | 0.33 | 0.29 |
| River | Difference in mean annual maximum temperature (℃) | | | | |
| SYR | 0.00 | 0.07 | 0.04 | 0.02 | 0.04 |
| CBR | 0.02 | 0.10 | -0.02 | 0.00 | 0.02 |
| HHR | 0.07 | 0.13 | 0.03 | 0.01 | 0.06 |
| FJR | 0.24 | 0.29 | 0.25 | 0.23 | 0.27 |
| | Difference in mean annual minimum temperature (℃) | | | | |
| SYR | -0.01 | 0.03 | 0.01 | -0.01 | 0.01 |
| CBR | -0.03 | 0.08 | -0.02 | -0.01 | 0.00 |
| HHR | 0.00 | 0.05 | -0.05 | -0.07 | -0.04 |
| FJR | 0.37 | 0.41 | 0.39 | 0.34 | 0.35 |
| River | Difference in mean annual precipitation (%) | | | | |
| SYR | 14.8 | 7.8 | 13.3 | 6.3 | 5.2 |
| CBR | 9.7 | 8.2 | 9.1 | 8.0 | 6.3 |
| HHR | 4.9 | 5.4 | 5.3 | 3.9 | 4.8 |
| FJR | 11.0 | 5.6 | 8.7 | 10.4 | 7.2 |

5    Table S3. Sensitivity results for pre-define parameters by SWAT for the four river basins

| Rank | SYR | CBR | HHR | FJR |
|---|---|---|---|---|
| 1 | ALPHA_BF | CN2 | CN2 | CN2 |

| 2 | GWQMN | ALPHA_BF | GWQMN | ESCO |
|---|---|---|---|---|
| 3 | TIMP | GW_DELAY | RCHRG_DP | SOL_AWC |
| 4 | CN2 | ESCO | ESCO | CANMX |
| 5 | SMTMP | GWQMN | SOL_AWC | GWQMN |
| 6 | SOL_AWC | CH_N | GW_REVAP | RCHRG_DP |

Table S4. Definition of identified sensitive parameters in SWAT hydrological model for the four river basins

| Parameters | Definition | Processes |
|---|---|---|
| ALPHA_BF | Baseflow recession constant (days) | Groundwater |
| CANMX | Maximum canopy storage (mm $H_2O$) | Runoff |
| CH_N | Manning coefficient value | Channel |
| CN2 | SCS runoff curve number for moisture condition II | Runoff |
| ESCO | Soil evaporation compensation factor | Evaporation |
| GW_DELAY | Delay time for aquifer recharge (days) | Groundwater |
| GW_REVAP | Groundwater "Revap" coefficient (days) | Groundwater |
| GWQMN | Threshold water level in shallow aquifer for base flow (mm) | Soil |
| RCHRG_DP | Deep aquifer percolation coefficient (fraction) | Groundwater |
| SMTMP | Threshold temperature for snow melt (℃) | Snow |
| SOL_AWC | Soil available water capacity (mm/mm soil) | Soil |
| TIMP | Snow temperature lag factor | Snow |

Table S5.The mean of middle-year of the 30-year samples for all GCMs under RCPs and under 1.5℃ or 2℃ global warming scenarios.

| threshold | RCP2.6 | RCP4.5 | RCP6.0 | RCP8.5 |
|---|---|---|---|---|
| 1.5℃ | 2029 | 2030 | 2032 | 2025 |
| 2.0℃ | × | 2049 | 2053 | 2038 |

Table S6. The agreements in simulated mean annual runoff and evapotranspiration based on WFD and downscaling climate simulation data from 5 GCMs for during 1961-2001 for the four river basins.

| River | GFDL-ESM2M | HadGEM2-ES | IPSL-CM5A-LR | MIROC-ESM-CHEM | NorESM1-M |
|---|---|---|---|---|---|
| | Difference in mean annual runoff (%) | | | | |
| SYR | 16.2 | 25.3 | 16.6 | 14.4 | 12.7 |
| CBR | -19.3 | 21.5 | 0.5 | -9.1 | -2.3 |

| River | | | | | |
|-------|------|-------|-----|------|------|
| HHR | -7.2 | 23.7 | 9.3 | 6.3 | 3.8 |
| FJR | -6.2 | -16.7 | 6.3 | 0.0 | -4.3 |
| River | Difference in mean annual evapotranspiration (%) | | | | |
| SYR | -3.3 | -37.7 | -4.7 | -19.8 | -17.6 |
| CBR | 12.4 | -0.5 | 6.6 | 7.5 | 3.2 |
| HHR | -1.8 | -8.1 | 0.8 | 3.2 | 4.8 |
| FJR | 15.5 | 13.4 | 4.7 | 11.7 | 12.7 |